# Chemogenomic analysis reveals key role for lysine acetylation in regulating Arc stability

Jasmin Lalonde [1,6], Surya A. Reis[1], Sudhir Sivakumaran[2], Carl S. Holland[1], Hendrik Wesseling[3], John F. Sauld[3], Begum Alural[1,4,5], Wen-Ning Zhao[1], Judith A. Steen[3] & Stephen J. Haggarty[1]

The role of Arc in synaptic plasticity and memory consolidation has been investigated for many years with recent evidence that defects in the expression or activity of this immediate-early gene may also contribute to the pathophysiology of brain disorders including schizophrenia and fragile X syndrome. These results bring forward the concept that reversing Arc abnormalities could provide an avenue to improve cognitive or neurological impairments in different disease contexts, but how to achieve this therapeutic objective has remained elusive. Here, we present results from a chemogenomic screen that probed a mechanistically diverse library of small molecules for modulators of BDNF-induced Arc expression in primary cortical neurons. This effort identified compounds with a range of influences on Arc, including promoting its acetylation—a previously uncharacterized post-translational modification of this protein. Together, our data provide insights into the control of Arc that could be targeted to harness neuroplasticity for clinical applications.

[1] Chemical Neurobiology Laboratory, Massachusetts General Hospital, Center for Genomic Medicine, Departments of Neurology & Psychiatry, Harvard Medical School, 185 Cambridge Street, Boston, MA 02114, USA. [2] Department of Anatomy and Neurobiology, Boston University School of Medicine, 72 East Concord, Boston, MA 02118, USA. [3] Boston Children's Hospital, F.M. Kirby Center for Neurobiology, Harvard Medical School, 3 Blackfan Circle, Boston, MA 02115, USA. [4] Department of Neuroscience, Institute of Health Sciences, Dokuz Eylul University, Izmir 35210, Turkey. [5] Izmir Biomedicine and Genome Center, Dokuz Eylul University, Izmir 35210, Turkey. [6] Present address: Department of Molecular and Cellular Biology, University of Guelph, 50 Stone Road East, Guelph, ON, Canada N1G 2W1. Correspondence and requests for materials should be addressed to J.L. (email: jlalon07@uoguelph.ca) or to S.J.H. (email: shaggarty@mgh.harvard.edu)

Neurons have a remarkable capacity to reorganize their structure, function, and connections in response to stimuli. These neuroplastic changes involve a diverse set of signaling molecules and protein effectors among which the activity-regulated cytoskeleton-associated protein (Arc, also known as Arg3.1) has come to be recognized as one of the central, and possibly most versatile, players[1,2].

Arc, whose expression is rapidly induced by signals coupled to neuronal activity, is a modular, "hub-like" protein[3] required for different forms of long-lasting synaptic plasticity including long-term potentiation (LTP)[4,5], long-term depression (LTD)[6], and homeostatic scaling[7]. At the synapse, it contributes to the endocytosis of 3-hydroxy-5-methyl-4-isoxazole receptors (AMPARs) by interacting with members of the endocytic vesicular machinery[8,9], influences the morphology of dendritic spines[10], as well as acts as a "tag" of inactive synapses[11]. Arc protein also accumulates in neuronal nuclei where it forms a complex with the nuclear spectrin isoform βSpIVΣ5 and acetyltransferase Tip60 to change

**Fig. 1** Profiling of signaling pathways supporting BDNF-induced Arc expression in mouse primary cortical neurons. **a** Western blots showing levels of Arc, phospho-p44/42 Mapk, and phospho-rpS6 in DIV13 primary cortical neurons that were treated for 6 h with a 4AP (40 μM) + Bic (50 μM) cocktail or BDNF (100 ng ml$^{-1}$). Untreated cells were included as control. **b–d** Graphs show mean ($n = 4$) Arc/β-actin (**b**), phospho-p42 Mapk/p42 Mapk (**c**), or phospho-rpS6/rpS6 (**d**) (±SEM) ratios for cells treated as in **a**. One-way analysis of variance (ANOVA) revealed a significant difference between treatment for each antigen (Arc, $F_{2,9} = 101.52$, $p < 0.0001$; phospho-p42 Mapk, $F_{2,9} = 31.74$, $p < 0.0001$; phospho-S6, $F_{2,9} = 34.49$, $p < 0.0001$). Tukey's HSD post hoc test, *$p < 0.05$; **$p < 0.01$; ****$p < 0.0001$. **e** Western blot of Arc protein expression in cortical neurons co-treated for 6 h with BDNF plus vehicle (DMSO) or Mek/Mapk pathway inhibitor U0126 (10 μM). **f** Similar experiment as in **e** was conducted with Mnk1 inhibitor CGP 57380 (5 μM) or Rac1 inhibitor NSC23766 (200 μM). Experiments in **e, f** were done in duplicate and lysates were probed for phospho-eIF4E, phospho-rpS6, and/or phospho-p44/42 Mapk (**g**) as control. **h** Arc mRNA expression in cortical neurons treated for 6 h with BDNF plus vehicle, U0126, or NSC23766. Inhibition of Mek/Mapk pathway by U0126 prevented BDNF-induced Arc expression but not Rac1 inhibition by NSC23766. Map2 expression was assessed as control. Bars represent mean fold-change measured by RT-qPCR (error bars indicate range from three biological replicates). **i** Schema summarizing the contribution of Mek/Mapk and Rac1/rpS6 signaling to BDNF-induced Arc expression in primary cortical neurons

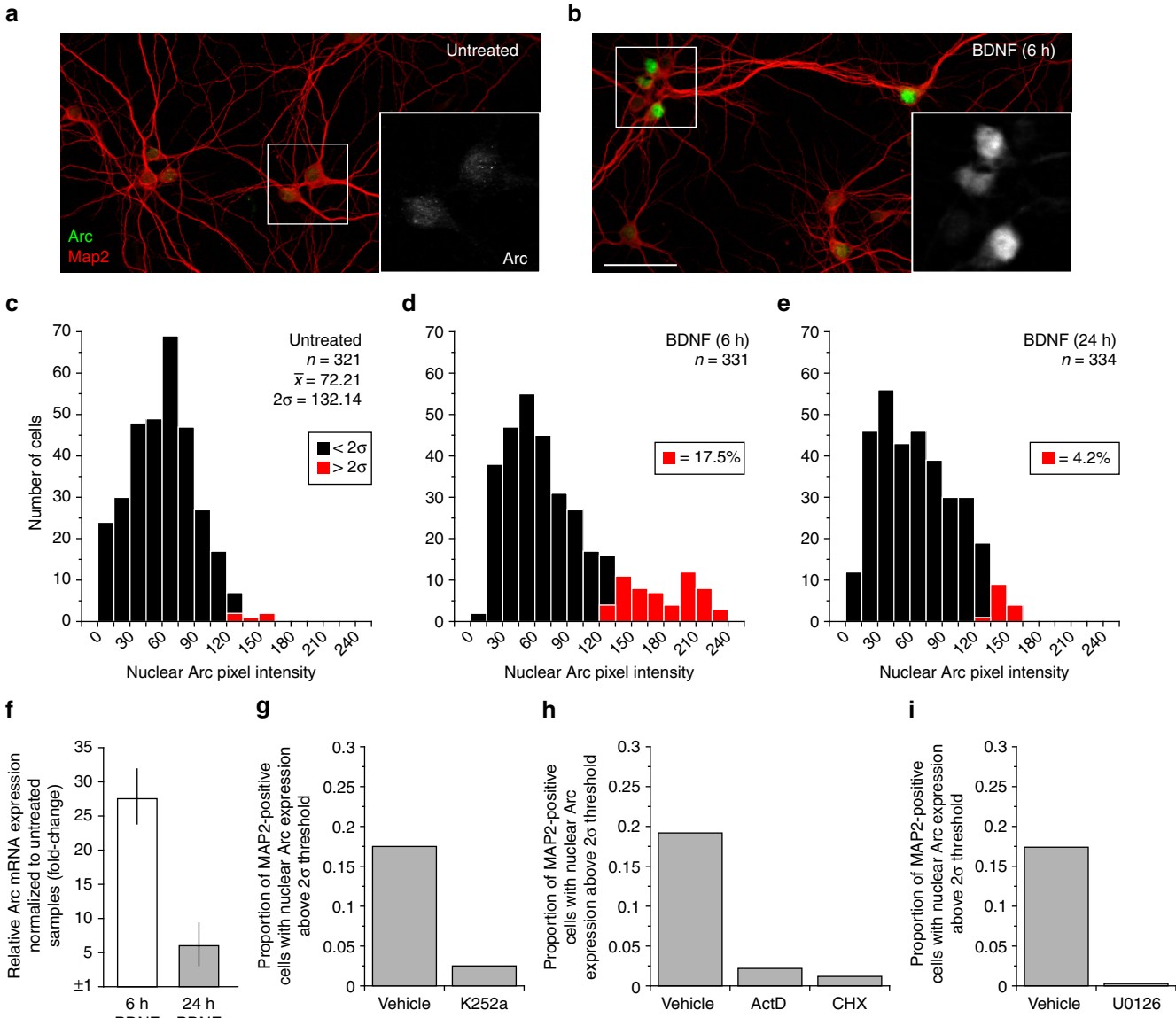

**Fig. 2** Validation of a high-content image-based nuclear Arc assay adapted for high-throughput chemical screening. **a**, **b** Representative immunostaining of Arc (green fluorophore) in untreated DIV14 mouse primary cortical neurons (**a**) and cells that were treated with BDNF for 6 h (**b**). Cells were co-immunostained with the neuronal marker Map2 (red fluorophore) to confirm specificity of staining to neurons. The high-magnification bottom-right insets in each panel show that Arc immunostaining in a BDNF-treated culture is particularly abundant in the nuclear compartment at this time point. Scale bar 50 μm. **c–e** Quantification of a set of biological replicates (triplicates) reveal that 17.5% of neurons (Map2-positive cells) have above threshold nuclear Arc expression (equal to or above 2σ of average nuclear Arc pixel intensity) 6 h after application of exogenous BDNF but only 4.2% after 24 h. **f** Quantitative real-time PCR Arc mRNA analysis shows a consistent difference as in **d**, **e** between cortical neurons treated for 6 and 24 h. Bars represent mean fold-change measured by RT-qPCR (error bars indicate range from three biological replicates). **g–i** BDNF-induced nuclear Arc protein expression is blocked by the co-application of a pan-inhibitor of receptor tyrosine kinases (k252a, 100 nM), RNA transcription (actinomysin D (ActD), 2 ng ml$^{-1}$), protein biosynthesis (cycloheximide (CHX), 50 μg ml$^{-1}$), or Mek/Mapk signaling (U0126, 10 μM), demonstrating that this assay can be adapted to perform a large-scale, image-based screening

the distribution of promyelocytic leukemia tumor suppressor protein nuclear bodies and the acetylation of histone H4K12[12,13]. Interestingly, nuclear Arc was implicated in decreasing cell-wide synaptic strength through repression of *GluA1* messenger RNA (mRNA) expression[14]. Together, these results illustrate the broad repertoire of molecular interactions and functions modulated and maintained by Arc in neurons.

A growing number of studies reveal connections between Arc and different neurodevelopmental disorders characterized by synaptic defects. For instance, the reduced level of AMPARs found at excitatory synapses in a mouse model of Angelman syndrome (*Ube3A* knockout) has been explained by a greater abundance of Arc protein resulting from the lack of Ube3A-mediated ubiquitination[15,16]. Likewise, impairments seen in the neurodegenerative disorder Gordon Holmes has been linked to loss-of-function mutations in another E3 ubiquitin ligase, Triad3A, which disrupts its capacity to target Arc protein for proteasomal degradation through ubiquitination[17]. Finally, enhanced mGluR-LTD in fragile X syndrome has been attributed to higher amount of Arc caused by reduced inhibitory control over its protein synthesis[18,19]. On the other hand, lower expression of Arc has been reported in the brain of Alzheimer's disease

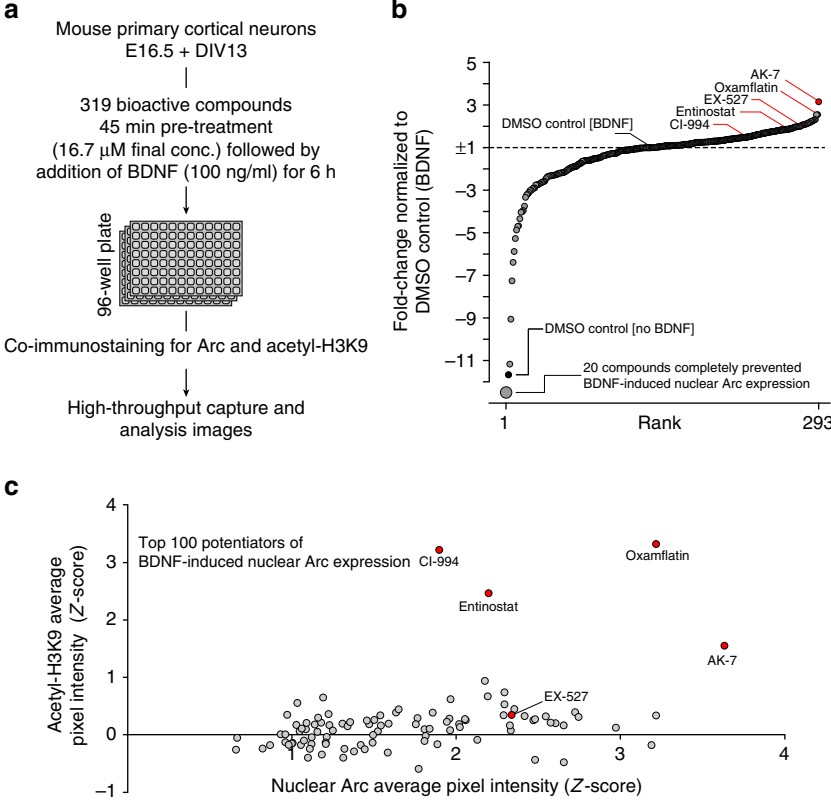

**Fig. 3** Screen for modulators of BDNF-induced Arc expression and histone acetylation. **a** Screen setup and technical details. **b** Waterfall plot of 293 screen compounds for which reliable data was collected at the tested concentration. Highlighted as red dots are KDAC inhibitor compounds that increased abundance of nuclear Arc. Data are presented as fold-change normalized to DMSO control plus BDNF (±1-fold-change, dashed line). **c** Joint representation as a Z-score of the acetyl-H3K9 and Arc immunofluorescence signals measured in neuronal nuclei for the top 100 potentiators of BDNF-induced Arc expression

(amyloid precursor protein overexpression) and Pitt-Hopkins syndrome (*Tcf4* haploinsufficiency) mouse models, but the causes and impact of this difference on synaptic function remains unknown in both cases[20,21]. Finally, Arc has been proposed to contribute to the pathophysiology of schizophrenia based on the observation that a significant number of schizophrenia risk genes identified by genome-wide association and sequencing studies are Arc binding partners[22,23]. Furthermore, *Arc* knockout mice have been reported to exhibit neurobehavioral and brain circuit alterations reminiscent of phenotypes seen in schizophrenia subjects[24]. Considering these findings as a whole, an intriguing question concerns whether reversing Arc abnormalities seen in these disease contexts could help alleviate cognitive and/or neurological impairments. However, limited knowledge of Arc regulation at the protein level, as well as what pharmacological strategies could be most effective to control its expression or function in physiological settings, have prevented full evaluation of this concept to date.

Here, we present results from a chemogenomic screen that searched for small molecules capable of enhancing Arc protein abundance or limiting its expression induced by brain-derived neurotrophic factor (BDNF) in primary mouse cortical neurons. Our effort led to the identification of multiple compounds with different structures and diverse targets. As we were validating a functionally related subgroup of the molecules that could strongly increase Arc expression in our screen, we recognized the potential for lysine acetylation to promote Arc protein stability. Using different biochemical strategies, including mass spectrometry, we validated this notion and identified specific lysine residues on Arc that can be either acetylated or ubiquitinated. These results

support a model where these two forms of post translational modification (PTM) compete with each other for specific sites to influence Arc abundance. Overall, our study reveals a new, unsuspected facet of Arc biology that may be targetable with pharmacological strategies in vivo to restore a normal balance of Arc protein in brain disorders characterized by its dysregulation.

## Results

**Design of an Arc assay adapted for chemogenomic screening.** As an initial step in the design of an assay that could assist with the discovery of pharmacological modulators of Arc, we tested whether recombinant BDNF treatment or the combined application of 4-aminopyridine (4AP, a blocker of $K_V1$ (Shaker, KCNA) family of voltage-activated $K^+$ channels) and bicuculline (Bic, a GABA receptor antagonist) caused a more robust expression of Arc protein in dissociated cortical neurons. As expected, western blot analysis revealed that cultures treated for 6 h with either BDNF or 4AP/Bic had both significantly more Arc protein than the control culture (i.e., untreated cells). However, application of BDNF also resulted in significantly higher Arc protein abundance than 4AP/Bic treatment (Fig. 1a, b). Difference between the two treatments could be attributed to the fact that BDNF, in comparison to 4AP/Bic, induced a significantly stronger phosphorylation of p44/42 mitogen-activated kinase (Mapk, also known as Erk1/2) and ribosomal protein S6 (rpS6) (Fig. 1), which are two key molecular effectors in pathways known to play a role in activity-dependent expression of Arc[1,2]. Importantly, we found that phosphorylation of Mapk and rpS6 induced by BDNF in our cortical neuron cultures preceded the

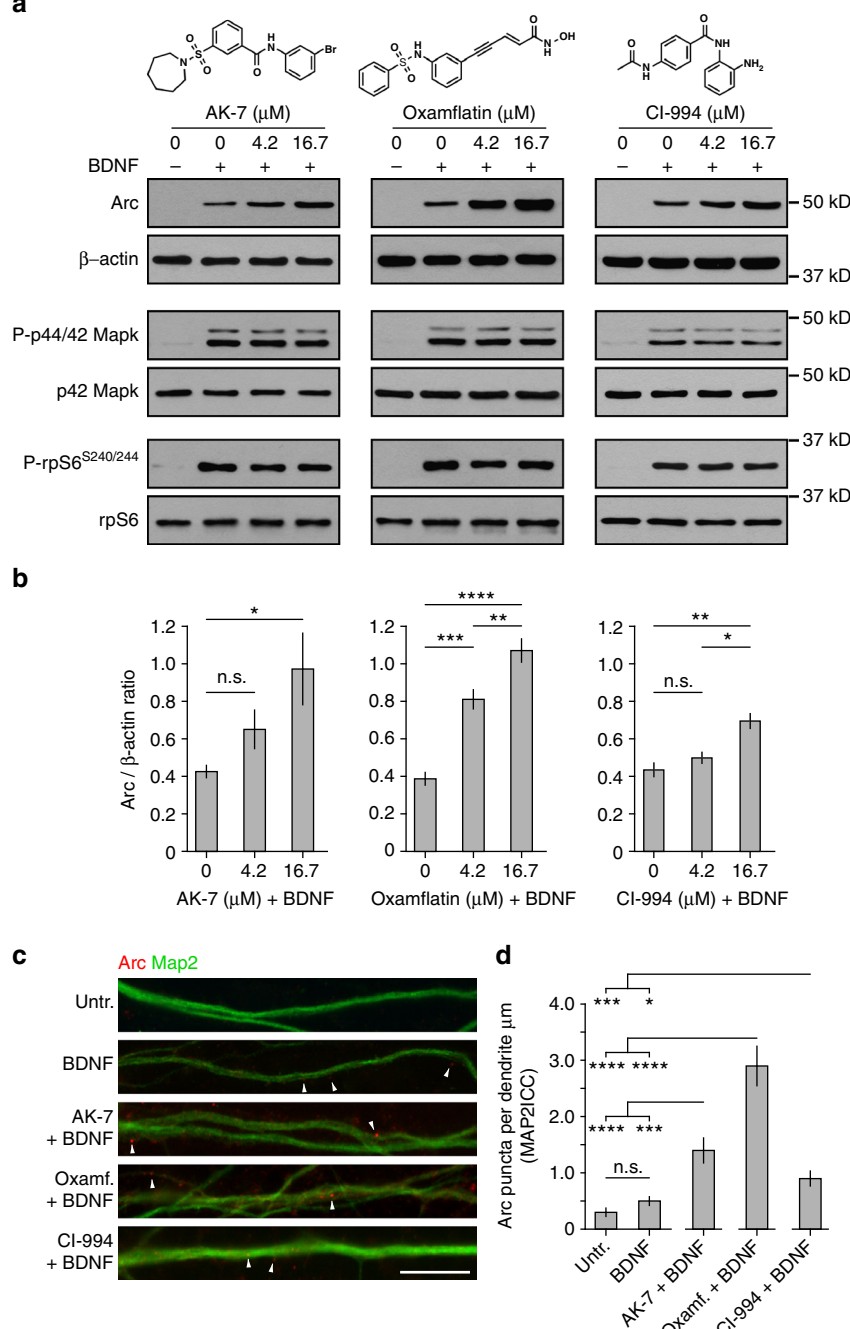

**Fig. 4** Post-screen validation of KDAC inhibitors AK-7, oxamflatin, and CI-994 potentiating effect on Arc protein abundance. **a** Western blot analysis reveals that lysates from BDNF-treated cortical cultures supplemented with AK-7, oxamflatin, or CI-994 have significantly more total Arc protein than BDNF plus vehicle (DMSO) control condition. Chemical structure of each compound is presented on top. **b** Graphs show mean ($n = 5$) Arc/$\beta$-actin ratio ($\pm$SEM) for cells treated as in **a**. One-way ANOVA revealed a significant dose–response difference between compound concentrations (AK-7, $F_{2,12} = 4.48$, $p < 0.05$; oxamflatin, $F_{2,12} = 46.72$, $p < 0.0001$; CI-994, $F_{2,12} = 12.24$, $p < 0.005$). Tukey's HSD post hoc test, *$p < 0.05$; **$p < 0.01$, ***$p < 0.005$; ****$p < 0.0001$. Western blots run with the same sample lysates revealed that these three KDAC inhibitors do not alter BDNF-dependent phosphorylation of p44/42 Mapk and rpS6. **c** Each tested KDAC inhibitor increases number of Arc puncta along dendrites labeled by Map2 immunocytochemistry. Representative captures of DIV14 mouse primary cortical neurons from each experimental condition co-immunostained for Arc (red fluorophore) and Map2 (green fluorophore). White arrowheads show examples of discrete Arc puncta. Primary cortical neurons were treated with BDNF and AK-7, oxamflatin, or CI-994 at a final concentration of 16.7 µM. Scale bar 10 µm. **d** Graph shows mean Arc puncta per dendrite µm for cells treated as in **c**. Separate ANOVA revealed a significant difference between cells co-treated with a KDAC and BDNF over untreated and BDNF alone conditions (AK-7, $F_{2,82} = 12.82$, $p < 0.0001$; oxamflatin, $F_{2,78} = 46.31$, $p < 0.0001$; CI-994, $F_{2,82} = 8.08$, $p < 0.001$). Tukey's HSD post hoc test, *$p < 0.05$; ***$p < 0.005$; ****$p < 0.0001$

gradual increase in Arc, as seen by western blotting (Supplementary Fig. 1). Furthermore, experiments using pharmacological inhibitors targeting Mek (U0126), Mnk1 (CGP 57380), and Rac1 (NSC23766), three critical factors in signal transduction pathways downstream of BDNF/TrkB[25–27], helped dissect the respective contribution of Mapk and rpS6 to BDNF-induced Arc expression in primary cortical neurons (Fig. 1e–h). Precisely, these results revealed that BDNF/TrkB signaling promotes the activity of both molecular cascades in a parallel but separate fashion such that Mek/Mapk signaling is essential for Arc transcription, while the activity of the Rac1/rpS6 pathway is exclusively important for Arc protein translation (Fig. 1i). Finally, although Mnk1 function has been reported to play a role in dentate gyrus BDNF/TrkB-induced Arc synthesis-dependent LTP[26,28], here in our ex vivo culture system of mouse cortical neurons we found that inhibition of this kinase with CGP 57380 did not prevent Arc expression despite impacting eukaryotic translation initiation factor 4E (eIF4E) phosphorylation as predicted (Fig. 1f).

Next, using fluorescent immunocytochemistry we assessed to what extent the observed induction of total Arc protein expression measured by western blotting corresponded to an accumulation of Arc in the nucleus of our cultured neurons. We chose to focus on nuclear Arc at this point based on the assumption that it would provide a robust and easily tractable phenotype for testing a large set of compounds with the help of a quantitative, single-cell level, high-content, image-based screen. Consistent with observations made by Korb et al.[14] in hippocampal neurons, we found that untreated cortical neurons (Map2-positive cells) displayed almost no evidence of nuclear Arc (Fig. 2a). On the other hand, many but not all of the neurons in cultures that were treated with BDNF for 6 h had high levels of Arc in their nuclei (Fig. 2b). To quantify this phenotype, we adopted an objective threshold (two standard deviations above the nuclear Arc immunofluorescence signal averaged from a representative population of untreated neurons) and found that 17.5% of the neurons ($n = 331$) from our 6 h BDNF-treated cultures had an Arc-specific average pixel intensity value above the established threshold in comparison with only 2.8% of the neurons ($n = 321$) in untreated cultures (Fig. 2c, d). In addition, we found that cultures treated for 24 h with BDNF had only 4.2% of the neuronal nuclei ($n = 334$) with above threshold average nuclear Arc signal (Fig. 2e). This result pattern was matched at the level of Arc mRNA expression, as measured by qRT-PCR (Fig. 2f). These results, in combination with the time-course analysis presented in Supplementary Fig. 1, led us to select the 6 h time point to conduct our screen. Finally, to further evaluate the sensitivity of our method to quantify nuclear Arc expression, we tested four pharmacological inhibitors predicted to prevent BDNF-induced Arc expression. As expected, analysis of high-resolution images of cortical neuron cultures co-treated with BDNF and a pan-inhibitor of receptor tyrosine and other kinases (K252a), RNA transcription (actinomycin D), protein translation (cycloheximide (CHX)), or Mek (U0126), shows that each of these inhibitors effectively prevented the increase of nuclear Arc in comparison to control cultures (Fig. 2g–i). These experimental results with known mechanism of action compounds critically guided the design and interpretation of the chemogenomic screen performed next.

**Report of nuclear Arc expression pharmacological modifiers.** To increase the throughput of our pharmacological screening for modulators of Arc, the above mentioned assay was adapted to cultures of mouse primary cortical neurons in 96-well plates. Using this assay, we screened a chemogenomic library according to the schematic shown in Fig. 3a. Importantly, we substituted

Map2 immunostaining for detection of acetyl-H3K9 in this part of our study because it provided a reliable nuclear marker for the automated analysis of nuclear Arc expression in cortical neurons and as a means of multiplexing the phenotypic analysis to include a measure of chromatin-mediated neuroplasticity[29,30]. This choice was supported by separate experiments that demonstrated specificity of the acetyl-H3K9 immunofluorescence signal to the Map2-positive cells in our primary cortical neuron cultures (Supplementary Fig. 2).

The chemogenomic library screened consisted of a custom collection of 319 compounds (Supplementary Data 1) with known or suspected activity within the nervous system, including an extensive selection of approved drugs used to manage symptoms associated with psychiatric disorders (antipsychotics, antidepressants, mood stabilizers, anxiolytics) as well as a large number of pharmacological agents with antiepileptic/anticonvulsant, neuroprotective, and/or nootropic (memory enhancing) properties in humans or pre-clinical animal models.

All 319 compounds were initially tested at a final concentration of 16.7 μM with three biological replicates. Analyzed results combining all three replicates are presented as a waterfall plot in Fig. 3b and as an annotated list in Supplementary Data 1. Of the 319 compounds screened in the library, high-resolution images were successfully collected from 293 compounds, with the remaining 26 compounds excluded from the data analysis because of either excessive cell death or confounding autofluorescence signal. As depicted in Fig. 3b, the effect of the 293 successfully tested compounds ranged from completely blocking to markedly increasing the prevalence of BDNF-induced nuclear Arc expression in reference to the control measure (i.e., DMSO control (BDNF)). In order to determine whether compounds with specific pharmacology clustered at one end or the other of this continuum, we surveyed the 40 most potent agents at each extremity of our data set. Interestingly, analysis of these two opposite groups revealed that the majority of the considered Arc "suppressors" were drugs with antipsychotic or antidepressant activity, while the majority of the Arc "potentiators" had neuroprotective and/or nootropic qualities (Supplementary Fig. 3). Although not all compounds in our library classified under these different pharmacological categories had an equivalent impact on BDNF-induced nuclear Arc expression, this observation suggests that our screen had identified subgroups of functionally related molecules affecting specific pathways implicated in the regulation of Arc.

Given that the number of known druggable targets for the treatment of cognitive impairments associated with psychiatric, neurological, and neurodevelopmental disorders is greatly limited, and that discovery of new lead candidates is needed to find more effective therapies, we considered that the connection that our screen uncovered between pharmacological agents with neuroprotective/nootropic qualities and nuclear Arc accumulation demanded further scrutiny. Consequently, we next focused our attention on the molecular and cellular impact of this specific set of compounds with the objective of understanding how they may exactly enhance BDNF-induced nuclear Arc expression. One intriguing class of neuroprotective/nootropic compounds that enhanced nuclear Arc levels in our screen were those annotated to be protein lysine deacetylase (KDAC) inhibitors[31,32]. Specifically, AK-7 and oxamflatin, two KDAC inhibitors with distinct chemical structures (Fig. 4a) and primary target affinity[33,34], were respectively the first and second strongest "potentiators". Furthermore, three other different KDAC inhibitors present in our library (EX-527, entinostat, and CI-994) also increased the occurrence of neuronal nuclei with above threshold nuclear Arc immunofluorescence signal by a factor of 1.5 or more in relation to the control measure (Fig. 3b; Supplementary Data 1). Since we

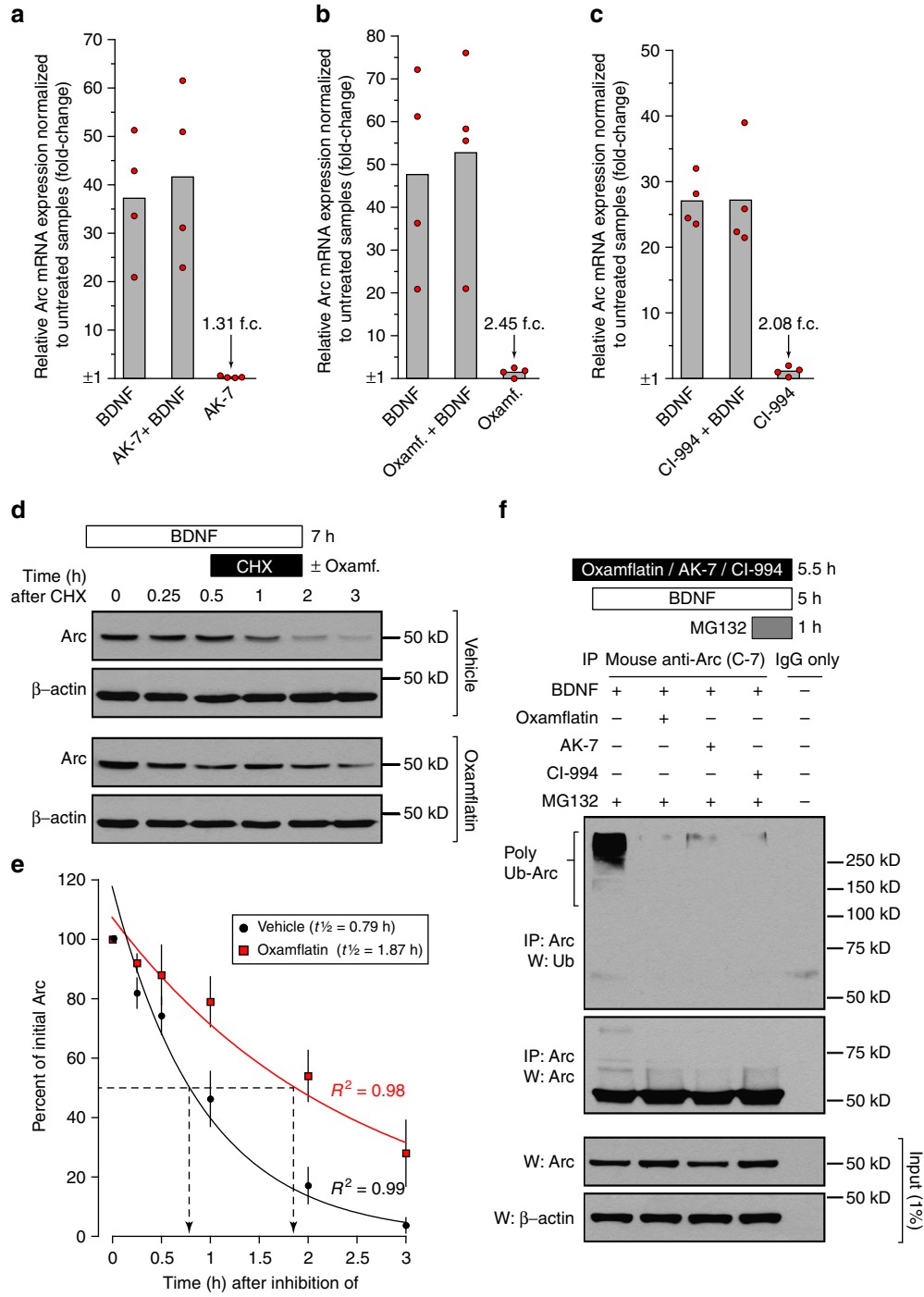

**Fig. 5** KDAC inhibitors AK-7, oxamflatin, and CI-994 influence Arc protein stability and ubiquitination but not transcription. **a–c** *Arc* mRNA levels were assessed by quantitative real-time PCR in primary cortical neuron cultures treated for 6 h with AK-7 (**a**), oxamflatin (**b**), or CI-994 (**c**). Each compound was tested at a concentration of 16.7 μM and in combination with BDNF or alone. *Arc* mRNA level induced by just BDNF was included in each experiment as a reference. Bars represent mean fold-change (f.c.) normalized to *Arc* expression measured in untreated cultures and each red point represents fold-change for an individual biological replicate ($n = 4$). **d** Primary cortical neurons were treated with BDNF for 4 h to induce Arc expression at which point protein synthesis was blocked by direct addition to culture media of cycloheximide (CHX, 50 μg ml⁻¹) in presence of vehicle or oxamflatin (16.7 μM). Lysates were collected at the indicated time points and Arc protein level in every sample was analyzed by western blot and normalized to corresponding β-actin. **e** Each data point from **d** is represented as mean (±SEM) calculated from five biological replicates and is presented as percentage of Arc level at the initial time point after CHX application ($t = 0$). Arc protein half-life was estimated by exponential decay curve fit. **f** Arc polyubiquitination was determined in lysates from primary cortical neurons cultured for 6 h with BDNF alone or BDNF plus oxamflatin, AK-7, or CI-994 (compounds final concentration = 16.7 μM). Lysates were immunoprecipitated with anti-Arc and western blotted with anti-Ub or anti-Arc as indicated. The proteasome inhibitor MG132 (10 μM) was added for 1 h before preparation of lysates to allow accumulation of polyubiquitinated Arc

had used acetyl-H3K9 immunostaining as a mask to facilitate the automated quantification of nuclear Arc protein expression in our screen, we leveraged this information in our data set to evaluate the activity of these five KDACs on this specific histone mark.

Joint representation as a Z-score of the acetyl-H3K9 and Arc immunofluorescence signals for each of the top 100 potentiators clearly illustrates the expected strong impact of the KDAC inhibitors oxamflatin, entinostat, and CI-994 on the level of H3K9

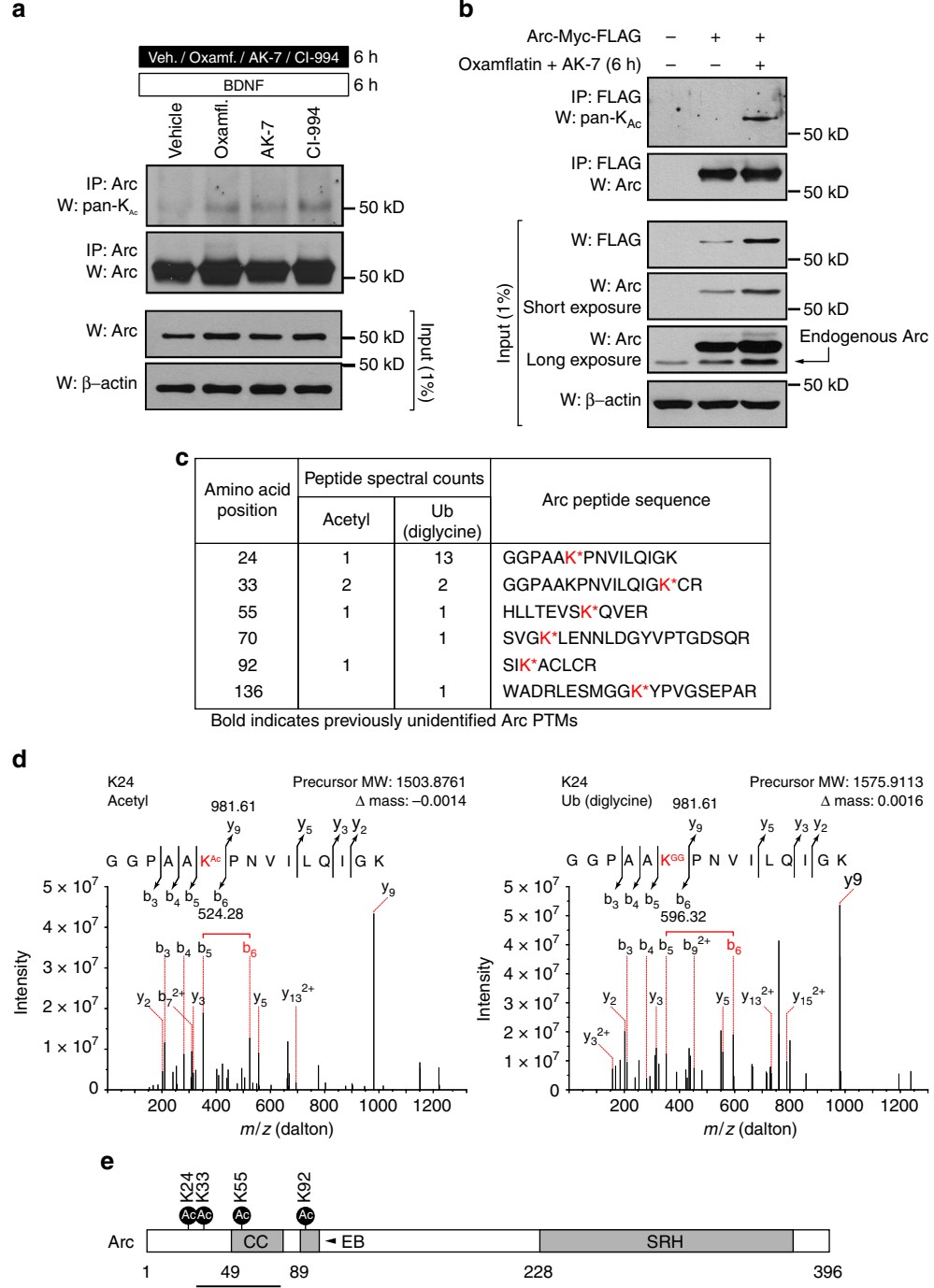

**Fig. 6** Arc is acetylated at different lysine residues. **a** Arc was immunopurified from lysates of primary cortical neurons co-treated for 6 h with BDNF and vehicle, AK-7, oxamflatin, or CI-994 as indicated. Immunoprecipitates were analyzed by western blot with an anti-acetyl-lysine antibody to reveal acetylated Arc. **b** Overexpressed mouse Arc-Myc-FLAG was immunopurified from Neuro2a cells treated as indicated. Immunoprecipitates were analyzed as in **a**. Acetylated Arc-Myc-FLAG was only detected from the Neuro2a cell lysate co-treated with oxamflatin (16.7 µM) and AK-7 (16.7 µM). **c** Table summarizes all acetylated and ubiquitinated (revealed as diglycine (GG) modification) lysine residues of Arc-Myc-FLAG immunopurified from overexpressing Neuro2a cells detected by MS/MS. The acetylated and/or ubiquitinated lysine residues in the isolated Arc tryptic peptides are presented in red (K*). **d** Representative product ion spectrum of MS/MS fragmentation supporting acetylation (left) and ubiquitination (right) of Arc lysine residue 24. **e** Schematic representation showing distribution of acetylated lysine residues over mouse Arc protein domains (CC coiled-coil, EB endophilin-3 binding region, NRD nuclear retention domain, SRH spectrin repeat homology)

acetylation (Fig. 3c). Interestingly, we also found with this analysis that AK-7 and EX-527, two molecules considered as "non-histone/ sirtuin-specific" KDAC inhibitors[35], had each a different impact on acetyl-H3K9 immunofluorescence. Precisely, AK-7 displayed an increasing effect of intermediate intensity on the acetyl-H3K9 mark, while EX-527 had no influence (Fig. 3c). In parallel, because of its structural similarities to the benzamide containing class I HDAC-selective inhibitor CI-994[31], we tested the inhibitory activity of AK-7 toward representative zinc-dependent KDACs within the family of class I, IIa, and IIb HDACs using an in vitro deacetylase biochemical assay. As shown in Supplementary Fig. 4, AK-7 failed to show any inhibitory activity toward the zinc-dependent KDACs, whereas in contrast, CI-994 exhibited potent inhibition of HDAC1/2/3, but not HDAC8 nor the class IIa HDAC5 or class IIB HDAC6. Overall, this mix pattern of influence on H3K9 acetylation led us to hypothesize that some of these KDAC inhibitors may be exerting their effect on BDNF-induced nuclear Arc expression through a mechanism uncoupled from effects on histone acetylation.

**Molecular basis of KDAC inhibitors influence on Arc**. To explore in greater detail the mechanistic basis of the hypothesis that effects on Arc levels could be disconnected from effects on histone acetylation, we then selected three of the top five KDAC inhibitors (AK-7, oxamflatin, and CI-994) representing three different chemotypes for further study. We first tested that the selected three KDAC inhibitors not only increased Arc in the nuclear compartment but really promoted accumulation of total Arc protein in cultured cortical neurons treated for 6 h with BDNF. Extending our screening results that focused only on nuclear Arc, western blot analysis showed that lysates from BDNF-treated cultures supplemented with either AK-7, oxamflatin, or CI-994 had overall significantly more Arc protein than the BDNF and vehicle control (untreated) culture (Fig. 4a, b). Importantly, separate western blots performed with the same sample lysates revealed that these three KDAC inhibitors did not alter BDNF-dependent phosphorylation of p44/42 Mapk and rpS6 (Fig. 4a), indicating that the increase in total Arc protein abundance caused by these compounds cannot be attributed to a greater activity of the signaling pathways upstream of these two factors (Fig. 1i). Next, to determine if the tested KDAC inhibitors were having an impact on the pool of Arc protein specifically found in dendrites, we used immunocytochemistry and high-resolution microscopy to quantify the abundance of Arc puncta along dendritic processes as revealed by Map2 co-immunostaining. This analysis found that co-treatment of AK-7, oxamflatin, or CI-994 with BDNF results in a significantly greater number of Arc puncta in dendritic projections of these samples than in those of untreated and BDNF alone conditions (Fig. 4c, d). Unexpectedly, we did not observe a significant difference between the untreated and BDNF alone conditions, which may be due to the duration/level of BDNF exposure, which was optimized for combination with KDAC inhibitors. Lastly, to complement these experiments, we also tested whether the potentiating effect of AK-7, oxamflatin, and CI-994 on BDNF-induced Arc protein expression could be explained by a change in the rate of *Arc* transcription or a decrease in *Arc* mRNA decay. We reasoned that if either one of these two alternatives was true, then we should expect to find a higher level of total *Arc* mRNA in BDNF-treated cultures supplemented with the selected KDAC inhibitors than in cultures treated with just BDNF. However, qRT-PCR analysis did not support this assumption for all three compounds (Fig. 5a–c), a surprising result since class I HDACs have been reported to repress BDNF-induced Arc expression in a recent paper[36]. Taken together, these results forced us to consider

the alternative mechanistic hypothesis that the KDAC inhibitor-dependent accumulation of Arc protein observed was mediated by changes in molecular events beyond *Arc* mRNA transcription and translation.

To directly test the hypothesis that the selected KDAC inhibitors increased Arc protein abundance by limiting its degradation, we calculated Arc protein half-life in primary cortical neuron cultures according to the experimental scheme depicted in Fig. 5d. Specifically, we stimulated the cells with BDNF for 4 h to promote accumulation of Arc protein and then added CHX to block all de novo protein translation. Importantly, cultures treated this way were divided in two groups—one to which oxamflatin was added at the same time as CHX and another that was treated with the vehicle only (baseline measure). A regression analysis of Arc protein abundance taking into account vehicle samples collected over a time-course of 3 h after the addition of CHX to the cells revealed a baseline half-life of ~47 min (Fig. 5e). Notably, this estimate of Arc protein half-life was almost identical to the one reported recently by Mabb et al.[37]. On the other hand, the regression analysis completed with oxamflatin-treated samples collected at similar time points after inhibition of protein synthesis revealed that addition of this KDAC inhibitor more than doubled Arc protein half-life (~112 min). This experiment provided clear evidence that the impact of oxamflatin on Arc protein abundance is, at least in part, the result of a decrease in the rate of Arc protein degradation; however, a mechanistic explanation of how this KDAC inhibitor is modulating the turnover of Arc protein remained elusive at this point.

To address this question, we then turned our attention to studies that found that lysine ubiquitination plays a major role in promoting Arc protein degradation by the proteasome[15,37,38]. These reports led us to hypothesize that if any of the selected KDAC inhibitors was enhancing Arc protein stability, then this molecular change might be also accompanied by a decrease in the level of Arc protein polyubiquitination. Hence, we immunopurified Arc from primary cortical neurons that were treated according to the design presented in Fig. 5f, performed western blotting with these samples, and probed the membrane with an ubiquitin antibody. This experiment confirmed our prediction that abundance of high-molecular weight polyubiquitinated Arc was sharply reduced by all three tested KDAC inhibitors when they were applied to the cultures (Fig. 5f). From this positive result, we then considered that the simplest model to reconcile how KDAC inhibitors could affect Arc ubiquitination would be analogous, for example, to how lysine acetylation of the microtubule-associated protein tau has been found to block its proteasomal degradation by acting as a barrier to the ubiquitination system[39]. Interestingly, in their study Min et al.[39] also found that inhibition of the KDAC sirtuin-1 with EX-527 was highly effective in promoting accumulation of acetylated tau—a result very similar to the observations that we made here about Arc and several KDAC inhibitors.

To the best of our knowledge, no evidence of Arc protein acetylation had been reported at the time of our studies. To determine if this was indeed the case, we immunopurified Arc from primary cortical neurons, separated these samples by SDS–polyacrylamide gel electrophoresis (PAGE), and probed the membrane with a pan-acetyl lysine antibody. This immuno-precipitation (IP)/western analysis suggested the presence of acetylated Arc in primary cortical neurons treated with either one of the selected KDAC inhibitors (Fig. 6a). To better support this observation, we overexpressed mouse Arc-Myc-FLAG in Neuro2a cells and repeated the IP/western analysis with our pan-acetyl lysine antibody. Consistent with data collected from primary cortical neurons treated with different KDAC inhibitors, a distinctive band corresponding to acetylated Arc-Myc-FLAG

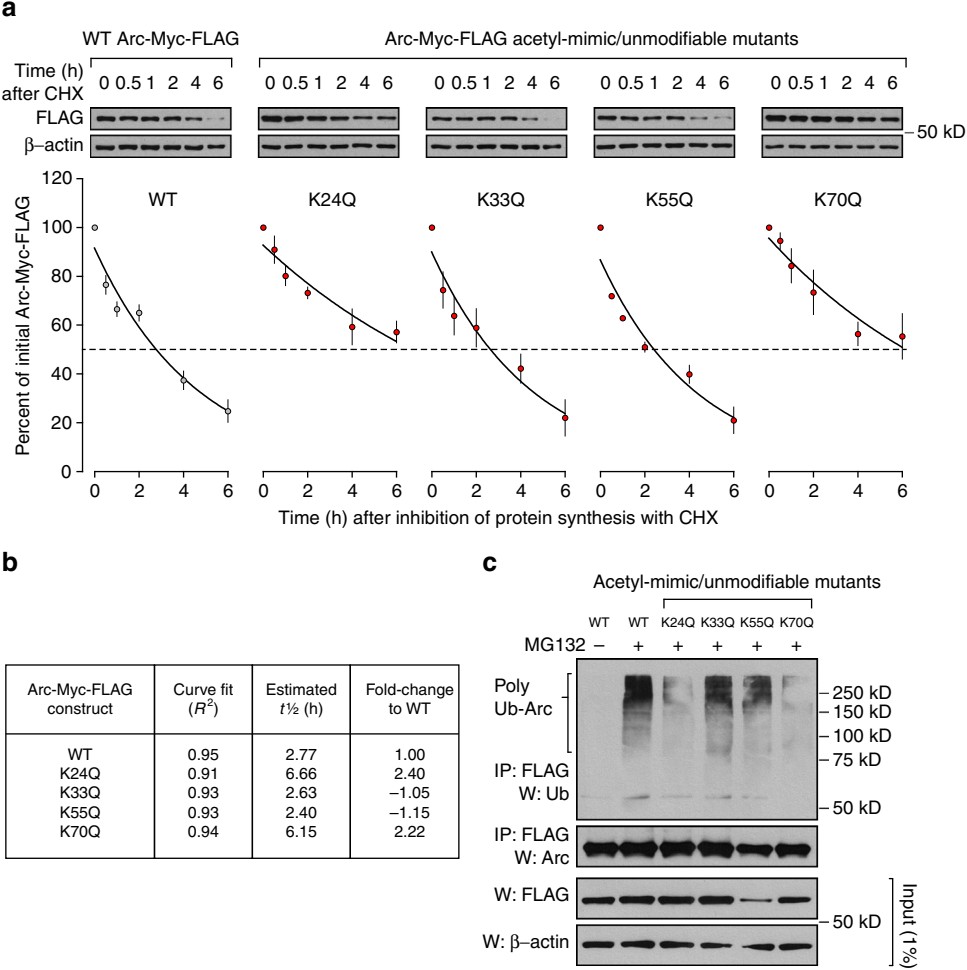

**Fig. 7** K24 and K70 contribute to Arc protein stability. **a** Mouse WT and acetyl-mimic mutants (K24Q, K33Q, K55Q, K70Q) Arc-Myc-FLAG were overexpressed in different Neuro2a cultures for 16 h at which point cells were treated with CHX (50 µg ml$^{-1}$) to block protein synthesis. Lysates were collected at the indicated time points and Arc-Myc-FLAG protein level analyzed in every sample by western blotting with an anti-FLAG antibody. Representative western blots are presented on top and quantification of data normalized to corresponding β-actin and presented as percentage of Arc-Myc-FLAG level (±SEM) at the initial time point after CHX application (t = 0). Curve represents exponential decay fit. **b** Estimate of protein half-life Arc for WT Arc-Myc-FLAG and each acetyl-mimic mutant is presented in table. **c** WT and acetyl-mutant Arc-Myc-FLAG polyubiquitination was determined in lysates from overexpressing Neuro2a cells. Lysates were immunoprecipitated with anti-FLAG and western blotted with anti-Ub or anti-Arc as indicated. The proteasome inhibitor MG132 (10 µM) was added for 20 min before preparation of lysates to allow accumulation of polyubiquitinated Arc-Myc-FLAG

was detected in lysate of cells dually treated with oxamflatin and AK-7 (Fig. 6b). We observed here that analysis of the input material for this experiment also suggested that joint application of the two KDAC inhibitors appeared to increase the abundance of both overexpressed and endogenous Arc in Neuro2a (Fig. 6b).

The fact that overexpressed Arc-Myc-FLAG could be acetylated in Neuro2a cells provided us the opportunity to next probe in an unbiased manner for the specific Arc lysine residue(s) modified. For this, we scaled-up immunopurification of Arc-Myc-FLAG overexpressed in Neuro2a cells and performed a mass spectrometric analysis with the intention of finding which lysine residues were specifically acetylated and/or ubiquitinated. These efforts allowed us to discover that these PTMs occur at several sites throughout Arc (Fig. 6c, d). Specifically, Arc tryptic peptides containing either an acetylation or ubiquitination (diglycine (GG)) mark at lysine residues 24, 33, and 55 were found, with each PTM detected in separate spectra. The three other sites were seen to be only acetylated (K92) or ubiquitinated (K70 and K136). Mapping of lysine residues that could be acetylated over protein domains suggests that these may have direct impact on how Arc interacts with other protein (Fig. 5e).

Next, we aimed to determine whether the K24, K33, K55, and K70 residues participate in Arc protein turnover. The first three sites were of particular interest since they could be either acetylated or ubiquitinated according to our mass spectrometric analysis. We also included K70 in this experiment because of its position in the coiled-coil and nuclear retention domains of Arc (Fig. 5e) as well as its relative proximity to K55. For this test, we used site-directed mutagenesis to generate plasmids that would allow overexpression of an acetyl-mimic/unmodifiable mutant Arc-Myc-FLAG protein (lysine (K) to glutamine (Q) mutant) for each of these four sites. After transfecting sets of Neuro2a cells with these different constructs, we inhibited translation with CHX and monitored overexpressed protein levels for up to 6 h after. Side-by-side comparisons revealed that K24Q and K70Q Arc-Myc-FLAG had a protein half-life that was more than two times higher than that measured for wild-type (WT) Arc-Myc-FLAG (Fig. 7a, b). Protein half-life estimates for K33Q and K55Q Arc-Myc-FLAG were essentially similar to WT. Further, we repeated this experiment with non-acetyl/unmodifiable Arc-Myc-FLAG protein (lysine (K) to arginine (R)) for the K24 and K70 residues and found, with these two other variants, an increase in protein

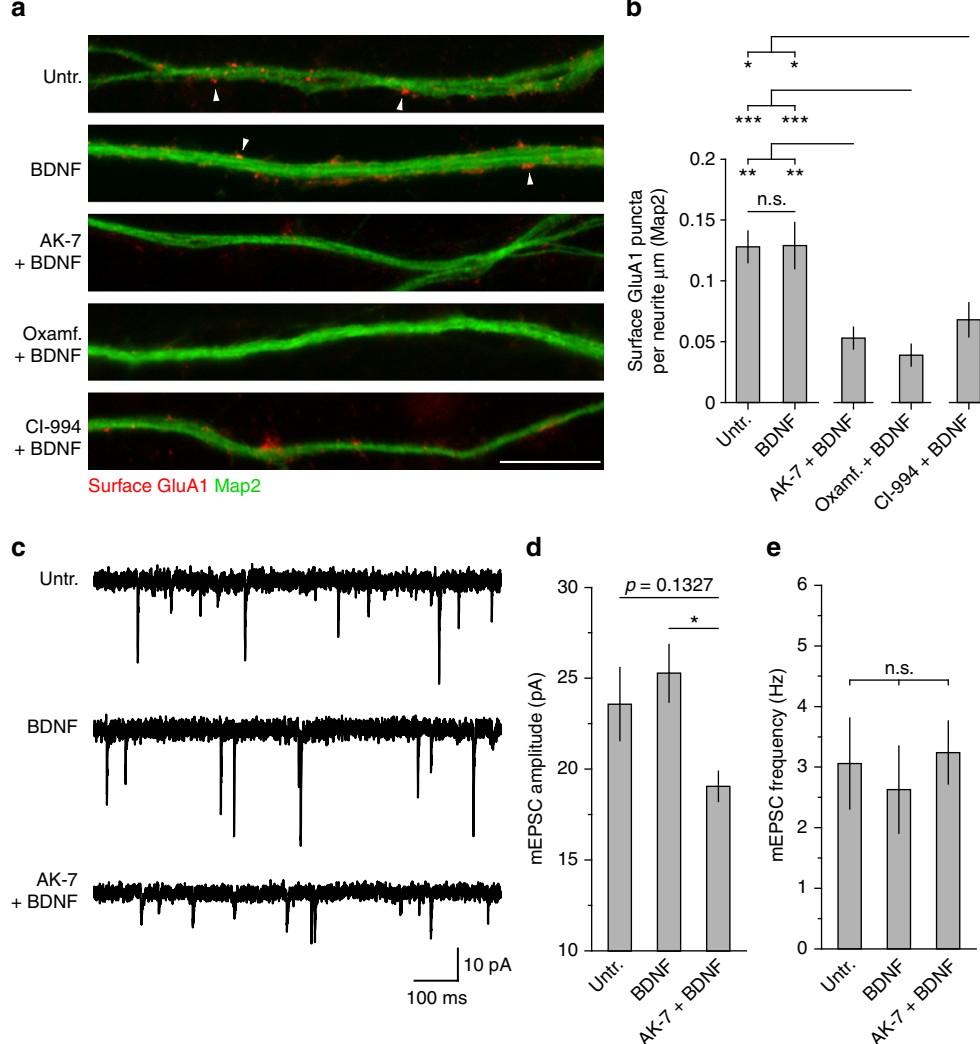

**Fig. 8** Effects of KDAC inhibitors AK-7, oxamflatin, and CI-994 on AMPAR biology. **a** Representative images of surface GluA1 immunostaining in DIV14 mouse primary cortical neurons for each tested experimental condition. AK-7, oxamflatin, or CI-994 was applied at a final concentration of 16.7 μM and treatment duration was 6 h with BDNF. Scale bar 10 μm. **b** Graph shows mean number of surface GluA1 puncta per dendrite μm for cells treated as in **a**. Separate ANOVA revealed a significant difference between cells co-treated with a KDAC inhibitor and BDNF over untreated and BDNF alone conditions (AK-7, $F_{2,72} = 7.28$, $p < 0.005$; oxamflatin, $F_{2,72} = 10.37$, $p < 0.0005$; CI-994, $F_{2,72} = 4.25$, $p < 0.05$). Tukey's HSD post hoc test, $*p < 0.05$; $**p < 0.01$; $***p < 0.005$. **c** Representative traces of mEPSCs from DIV15 mouse primary cortical neurons treated according to the indicated treatment (untreated, BDNF alone, or AK-7 plus BDNF). Cells that were co-treated with AK-7 and BDNF for 6 h prior to recording show lower mEPSC amplitude. **d** Quantification of mEPSC amplitude for experimental conditions shown in **c**. One-way ANOVA revealed a significant difference in mEPSC amplitude between the BDNF alone and AK-7 plus BDNF condition ($F_{2,22} = 3.89$, $p < 0.05$). Tukey's HSD post hoc test, $*p < 0.05$. A total of 5086 (untreated, $n = 9$ cells), 3760 (BDNF alone, $n = 8$ cells), and 4475 (AK-7 plus BDNF, $n = 8$ cells) mEPSC events were analyzed from three independent cultures. **e** Quantification of mEPSCs frequency shows no differences between experimental conditions

half-life comparable to the one measured for their acetyl-mimic/ unmodifiable protein counterpart (Supplementary Fig. 5). This result provides, in the case of K24 specifically, supporting evidence to the idea that a direct competition between acetylation and ubiquitination can occur at a precise residue to change Arc protein stability and abundance. Finally, to test if the greater stability observed for K24Q and K70Q Arc-Myc-FLAG could be resulting from a lower rate of polyubiquitination, we performed an IP/western assay and found, as expected, a noticeable decrease in polyubiquitination of these two mutants but not K33 or K55. Based on these results, we propose that: (1) Arc protein can be acetylated and that this PTM prevents the rapid degradation of Arc by competing with the ubiquitination system, (2) KDACs appear to be actively reversing Arc acetylation in order to promote its degradation, but that this process can be interfered with by the KDAC inhibitors oxamflatin, AK-7, or CI-994, and 3) that interfering with Arc deacetylation in cortical neurons as well as Neuro2a cells leads to its accumulation.

**Consistent effects of KDAC inhibitors on AMPAR biology.** We next sought to understand the physiological consequence of modulating Arc stability through reversible control of its acetylation state. In particular, Arc has been shown to impact AMPAR-mediated synaptic transmission in different ways like at individual post-synaptic terminals where it can facilitate AMPAR internalization in response to neuronal activity by recruiting the endocytosis effectors endophilin, dynamin, and clathrin-adaptor protein 2[8,9,40]. In neuronal nuclei, Arc can also contribute to repression of *GluA1* transcription, which affects cell-wide

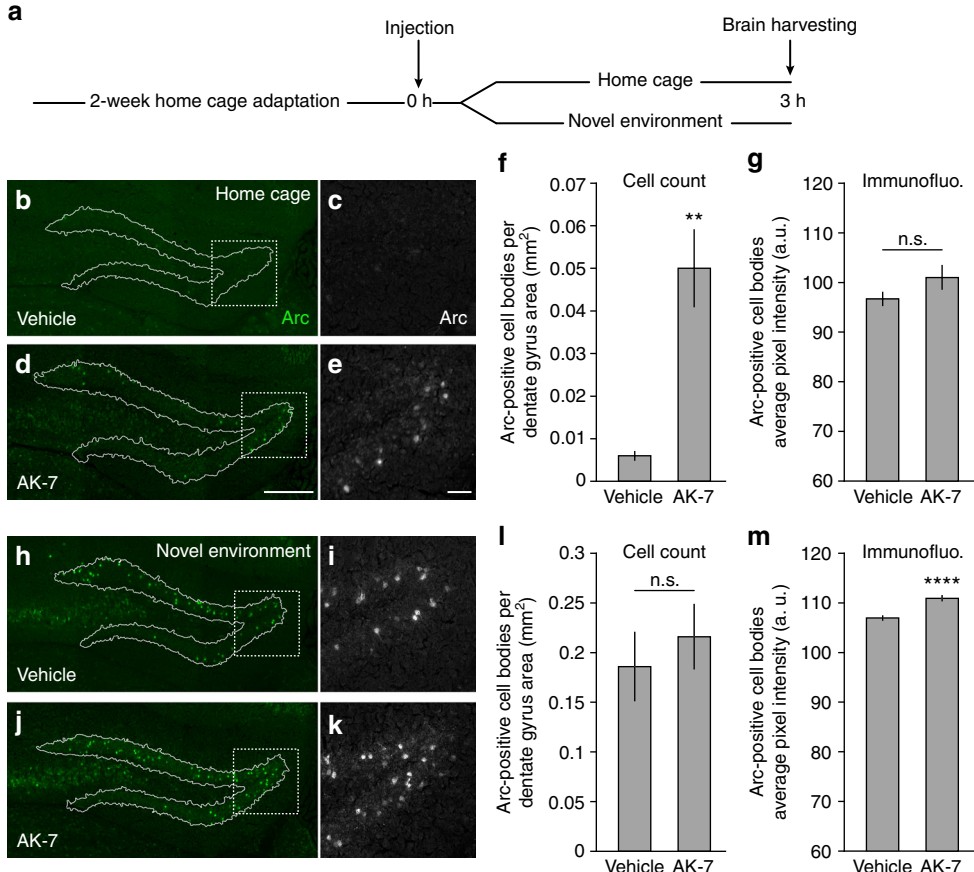

**Fig. 9** KDAC inhibitor AK-7 influences Arc protein level in vivo. **a** Experimental design used to assess the influence of AK-7 on Arc protein expression in mice dentate gyrus exposed to a familiar (home cage) or novel environment. **b**–**e** Representative captures of Arc immunostaining in the dentate gyrus of mice 3 h after intraperitoneal injection of vehicle (**b**, **c**) or AK-7 (**d**, **e**, 30 mg kg$^{-1}$). Scale bar 250 μm. Mice were returned to their home cage immediately after the injection and left undisturbed for the time period before killing. High-magnification image of the area highlighted by a dotted square in **b**, **d** are presented in **c**, **e**, respectively. Scale bar 50 μm. **f**, **g** Quantification of number of immunopositive Arc cells per mm$^2$ for the whole dentate gyrus of mice injected with vehicle or AK-7 and returned to their home cage (**f**) and average Arc immunofluorescence signal intensity in cell bodies (**g**). Data represent average made from three mice for each experimental condition. **$p < 0.01$, two-tailed $t$-test. **h**–**k** Representative captures of Arc immunostaining in the dentate gyrus of mice treated as in **b**–**e** but that were placed in a novel environment for 3 h immediately after the vehicle (**h**, **i**) or AK-7 (**j**, **k**) injection. Strong Arc expression caused by novel environment exposure can be observed in both conditions. **l**, **m** Quantification of number of Arc immunopositive cells per mm$^2$ in dentate gyrus reveals that no difference between the two injection groups for mice exposed to a novel environment ($p = 0.571$, two-tailed $t$-test) but quantification of Arc immunofluorescence in cell bodies was significant. ****$p < 0.0001$, two-tailed $t$-test. Data represent average made from three mice for each experimental condition. Dentate gyrus boundaries in each tissue section (with line) was segmented according to DAPI staining

AMPAR-mediated transmission in this case[14]. Together, these results led us to ask whether promoting Arc protein acetylation and stability with the different KDAC inhibitors that we characterized could modify AMPAR biology in a manner consistent with the observed increase in Arc protein. To answer this question, we first tested the influence of AK-7, oxamflatin, and CI-994 on the occurrence of surface GluA1 in primary cortical neurons treated with BDNF for 6 h. As shown in the first two top panels of Fig. 8a, immunostaining under non-permeabilizing conditions of the AMPAR subunit GluA1 with an antibody recognizing the protein extracellular N-terminus region reveals the presence of clear, distinct puncta along dendrites of cortical neuron cultures that were either untreated (baseline control) or exposed to exogenous BDNF alone. Note here that BDNF application did not influence, in one way or the other, the number of GluA1 puncta in comparison to the baseline condition (Fig. 8b). In contrast, cultures that were subjected to the co-application of BDNF and AK-7, oxamflatin, or CI-994 displayed a significant decrease in surface GluA1 levels (Fig. 8a, b). These results were entirely consistent with our observations regarding dendritic Arc puncta

described above where we observed no difference between untreated and BDNF alone conditions in the number of Arc puncta as well, but a significant increase in samples that were co-treated with either one of the three KDAC inhibitors and BDNF (Fig. 4c, d). Taken together, these two experiments suggest that the local accumulation of Arc in dendrites of cells treated with KDAC inhibitors and BDNF may be promoting widespread AMPAR internalization through molecular mechanisms that described in previous reports[8,9,40]. To support the idea that local molecular events in dendrites is affecting surface GluA1 abundance, we conducted a qRT-PCR analysis and found no evidence that the reduction in surface levels of GluA1 was explained in the case of the AK-7 condition by the transcriptional regulation of this AMPAR subunit (Supplementary Fig. 6). Finally, to complement this group of functional experiments we performed single-cell electrophysiological recordings to assess amplitude and frequency of AMPA-mediated miniature excitatory post-synaptic currents (mEPSCs) between untreated, BDNF alone, and cultures co-treated with AK-7 and BDNF. As expected, and consistent with our GluA1 and Arc imaging data, we found a significant

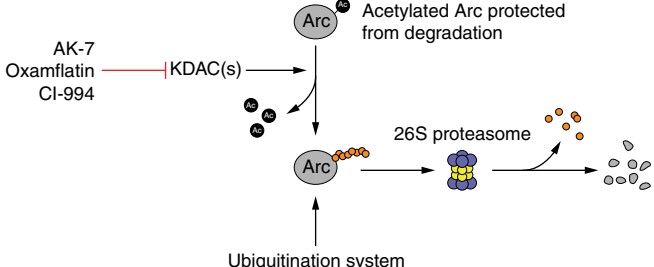

**Fig. 10** Model of dynamic interplay between lysine acetylation and ubiquitination in the control of Arc protein stability and proteasomal degradation. Hypothetical model of how lysine acetylation could influence the polyubiquitination of Arc protein and what influence some of the KDAC inhibitors identified in our chemogenomic screen could exert on this process

decrease in the amplitude of mEPSCs for cells co-treated with AK-7 and BDNF (Fig. 8c, d) compared to BDNF alone. For its part, mEPSC frequencies between the different conditions were all similar (Fig. 8e). In sum, these results provide converging evidence that pharmacological strategies that affect Arc protein acetylation and stability in a direct fashion can be leveraged to manipulate cellular processes susceptible to Arc function, such as AMPAR-mediated synaptic transmission.

**Arc acetylation may be pharmacologically targetable in vivo.** Our chemogenomic screen identified a range of pharmacological agents that can modulate BDNF-induced Arc expression in primary cortical neurons. Among those drugs, a number of KDAC inhibitors displayed the strongest tendency to increase Arc abundance, which we propose occurs because these compounds are preventing Arc lysine deacetylation and, consequently, blocking the access of these residues to the ubiquitination system. Whether Arc levels can be influenced in vivo through this mechanism represents a question that could have important implications from a brain disorder therapeutic perspective. Consequently, we conducted an experiment that tested if systemic administration through intraperitoneal injection of the KDAC inhibitor AK-7 could enhance Arc protein abundance in the mouse dentate gyrus—a region of the hippocampus that has very low Arc protein level in unstimulated animals but shows high induction when animals are exposed to a novel environment[41]. We chose to test the AK-7 compound here because it was the strongest potentiator of BDNF-induced Arc expression in our screen and a molecule that can efficiently cross the blood–brain barrier according to previous reports[34,42]. As expected, cell count of Arc-positive cell bodies in the dentate gyrus of animals that were injected with the vehicle solution and immediately returned to their home cage for 3 h was very low (Fig. 9b, c). However, animals that were exposed to the same context but were instead injected with a single dose ($30\,mg\,kg^{-1}$) of AK-7 were found to have more than eight times the number of neurons with prominent Arc immunostaining (Fig. 9d–f). However, comparison of the immunofluorescence signal intensity for the cells expressing Arc in these two groups was not statistically different (Fig. 9g). Further, as evidence that intraperitoneal injection of AK-7 could reach neurons, we also measured acetyl-H3K9 immunofluorescence in adjacent tissue sections of each animal and report a small but statistically significant increase in staining intensity (Supplementary Fig. 7). This result matches the effect of this compound on this histone mark that we have quantified earlier in our screen (Fig. 3c). To finish, we also tested whether AK-7 injection could enhance Arc levels under circumstances where it is already highly expressed. As seen in Fig. 9h and i, mice injected

with the vehicle solution and then introduced to a novel environment with new objects for 3 h results in a strong Arc expression throughout the dentate gyrus. Repeating the same procedure but with mice injected with AK-7 did not led to more Arc immunopositive cell bodies but analysis of immunofluorescence intensity was statistically significant (Fig. 9j, l). Together, these results support the idea that Arc protein acetylation could be directly targeted with pharmacology in vivo to modulate Arc protein abundance.

## Discussion
An explicit goal of translational neuroscience is to find strategies capable of exploiting the inherent ability of the nervous system to undergo neuroplastic changes and to adapt this knowledge for the treatment of brain disorders[43]. Our study directly connects to this mission by systematically searching for modulators of Arc biology, one of the most recognized regulators of neuroplasticity[1,2,44]. Growing evidence supports the potential therapeutic value of targeting Arc under certain disease contexts. For instance, genetically lowering Arc expression has been shown to ameliorate seizure-like phenotypes in an Angelman syndrome mouse model[45], suggesting that achieving the same through neuropharmacology could offer comparable benefits. Another example involves nuclear Arc specifically, which has been found to sustain a homeostatic process that limits cocaine-induced chromatin remodeling and long-term behavioral changes[46]. Based on their findings, the authors of this study infer that modulating Arc could help in the treatment of drug addiction. Finally, as mentioned earlier tantalizing results provided by genetic association studies[22,23] and the detailed phenotyping of the Arc knockout mouse model[24] position Arc as a possible contributor to the pathophysiology of schizophrenia. Despite the fact that a direct connection between Arc and schizophrenia remains to be found, the recent identification of the Arc N-terminal lobe as a critical domain for intermolecular binding provided further support to the idea that rare variants in genes encoding synaptic proteins could alter their interaction with Arc resulting in disruption of cellular processes controlling neuronal excitability[47]. It is highly likely that connections between Arc and different brain disorders will continue to emerge in the near future with further proteomic studies of the Arc complex and expanded human genetic studies. As this unfolds, there will be a concomitant growing need to test in more detail whether manipulating the expression or function of this key regulator of neuroplasticity provides any therapeutic gains. Given the positive imaging and electrophysiological results related to AMPAR-mediated transmission that we present in this study for the three KDAC inhibitors tested, this seems readily testable (Fig. 8). In sum, the screening strategy and extended list of Arc modulators in this study will represent valuable resources with potential for repurposing of approved drug once further pre-clinical translational work is performed.

The chemogenomic library that we have tested largely consisted of molecules with well-established cellular targets and clinical uses. This allowed us to classify the drugs that most strongly impacted BDNF-induced nuclear Arc expression, in one way or another, according to specific categories (Supplementary Fig. 3). Doing so, we observed at one end of the spectrum that a large number of antipsychotic and antidepressant drugs in our library acted to prevent Arc expression. Perhaps the most likely explanation for this specific set of molecules consists in alteration of the Mek/Mapk and/or rpS6 pathways as a result of acutely antagonizing dopaminergic signaling with antipsychotic drugs or enhancing serotonergic activity with antidepressants[48,49]. More specifically, since Mek/Mapk and rpS6 activities are critically

needed for BDNF-induced Arc expression, as the work of other groups[1,2] and ours have shown (Fig. 1), it should come as no surprise that any treatment interfering with these cascades will cause an attenuation of Arc induction. Then, considering the other end of our data set where BDNF-induced Arc expression was potentiated, we found a less-well-defined group of small molecules with diverse reported targets. Nevertheless, as we categorized this subset it became clear that: (1) connections to neuroprotective and/or nootropic effects could be made for many of these compounds, and (2) that several agents displayed inhibitory activity toward different KDACs. By trying to understand how these different KDAC inhibitors without pronounced effects on Arc transcription or on histone acetylation could act to promote Arc protein abundance that we were led to discover, unexpectedly, the interplay between lysine acetylation and ubiquitination in the control of Arc protein stability and accumulation. A model summarizing these interactions is presented in Fig. 10.

An open question concerning the neurobiology of Arc is how this single protein can contribute to various forms of neuroplasticity in response to different stimuli and on different time scales. The complete explanation to this question will likely include the revelation of an intricate set of protein PTMs that act to confer different molecular and functional specificity to Arc as well as potential subcellular distribution into distinct protein complexes. To date, only a small number of reports have provided evidence for the influence of PTMs, which includes ubiquitination[15,37], sumoylation[50,51], as well as phosphorylation[52,53], on Arc biochemistry and function. In this study, we uncovered a role for lysine acetylation in the control of Arc protein stability. On this note, the existence of regulatory crosstalk between lysine acetylation and ubiquitination has been documented for a number of proteins[39,54]; however, the implication of this link for the regulation of proteins critical to neuroplasticity has seen little attention. Interestingly, it has been estimated through proteome-wide analysis that about one-third of lysine acetylation sites in human cells are also subject to ubiquitination[55]. Hence, our finding of this competition between these two forms of lysine PTM for Arc cannot be considered an isolated observation but rather a mechanism that is often seen with proteins that have a rapid turnover rate.

Finally, an additional open question concerns the molecular underpinnings of reversible Arc acetylation and deacetylation. This topic will require systematic analysis to identify the implicated enzymes in this process but hypotheses supported by converging evidence can be made at this point. One acetyltransferase candidate could be Tip60, for example, which has already been reported to form a complex with Arc in neuronal nuclei in order to acetylate histone H4K12[13]. Since Arc can associate with different components of the cytoskeleton[56,57], another possibility worth considering could be α-tubulin acetyltransferase 1 (Atat1), which is the enzyme that acetylates α-tubulin[58]. As for the relevant deacetylases, the substrate specificity of oxamflatin and AK-7, the two KDAC inhibitors that had the strongest effect at potentiating Arc in our screen, certainly represent a valid starting point to investigate this question. Whereas oxamflatin is known as a broadly acting inhibitor of class I (HDAC1/2/3/8) and class IIb (HDAC6) KDACs, which overlaps with the reported selectivity of CI-994 toward class I (HDAC1/2/3) KDACs, AK-7 is particularly interesting as it is considered a sirtuin-2-specific inhibitor. Our studies confirmed its lack of inhibition of class I/IIb HDACs. Relating sirtuin-2 to Arc biology would be an exciting result as inhibition of this KDAC by AK-7 has been found to prevent neurodegeneration in mouse models of Huntington's, Parkinson's, and Alzheimer's diseases[40,59,60]. Whether any of these positive effects of AK-7 on

neuronal survival and cognition occur through change in Arc protein abundance or function due to a decrease of its deacetylation would be a finding of utmost importance.

## Methods

**Cell culture and transfection.** Forebrains of E16.5 C57BL/6 mouse embryos were dissected and then dissociated in trypsin solution for 15 min followed by three washes with phosphate-buffered saline (PBS). Single-cell suspension was achieved by gentle pipet trituration. Neurons were seeded in poly-L-lysine/laminin-coated six-well plates at a density of $2 \times 10^6$ per well and maintained in neurobasal medium containing B27 supplement (2%, Invitrogen, Grand Island, NY), penicillin (50 units ml$^{-1}$, Invitrogen), streptomycin (50 µg ml$^{-1}$, Invitrogen), and glutamine (1 mM, Sigma). For experiments involving BDNF (EMD Millipore Corps, Billerica, MA), the stock solution (2 ng µl$^{-1}$ in water) was added directly to the culture medium at a final concentration of 100 ng ml$^{-1}$ for the indicated period of time. Preparation of mouse primary forebrain cortical neuron cultures was approved by the Animal Care and Use Committee of the Massachusetts General Hospital and carried out according to institutional guidelines.

Neuro2A cells were cultured in DMEM (supplemented with 10% fetal bovine serum (Gemini-Bio-Products, West Sacramento, CA), penicillin (50 units ml$^{-1}$), and streptomycin (50 µg ml$^{-1}$)) and transfected overnight using Lipofectamine 2000 (Invitrogen) according to the manufacturer's protocol. The Neuro2A cell line was generously shared by Dr. Grace Gill.

**Antibodies and pharmacological compounds.** The antibody recognizing Arc (#156 005, 1:2000 for WB and 1:1000 for IF) was purchased from Synaptic Systems (Goettingen, Germany). The specificity of this antibody was validated using immunizing peptide pre-incubation and IP/western assay (Supplementary Fig. 8). The antibodies recognizing Arc (sc-17830, IP), p42 Mapk (Erk2, sc-1647, 1:1000 for WB), ubiquitin (sc-8017, 1:200 for WB), and horseradish peroxidase-conjugated secondary antibodies were from Santa Cruz Biotechnology (Santa Cruz, CA). The antibodies recognizing phosphorylated p44/42 Mapk (Erk1/2$^{Thr202/}$$^{Tyr204}$, #4370, 1:1000 for WB), rpS6 (#2317, 1:400 for WB), phosphorylated rpS6$^{Ser240/244}$ (#2215, 1:1000 for WB), eIF4E (#9742, 1:500 for WB), phosphorylated eIF4E$^{Ser209}$ (#9741, 1:500 for WB), H3 (#3638, 1:1000 for WB) as well as acetylated lysine (#6952, 1:1000 for WB) were acquired from Cell Signaling Technology (Beverly, MA). Finally, the antibodies recognizing acetylated-H3K9 (#07-352, 1:5000 for WB and IF), GluA1 (MAB2263, 5.0 µg ml$^{-1}$ for surface imaging), and Map2 (AB5543, 1:1000 for IF) were purchased from EMD Millipore Corp, while the β-actin (A1978, 1:10,000 for WB) and M2 FLAG (F1804, 1:1000 for WB) antibodies were from Sigma-Aldrich (St-Louis, MO).

Actinomycin D, oxamflatin, and AP5 were from Santa Cruz Biotechnology. AK-7, 4AP, bicuculline, CGP 57380, and tetrodotoxin were from Tocris Bioscience (Bristol, UK). Cycloheximide, MG132, NSC23766, and U0126 were from EMD Biosciences (La Jolla, CA). K252a was from Sigma-Aldrich and CI-994 was synthesized at the Broad Institute (Cambridge, MA). The chemogenomic library (Selected Molecular Agents for Rett Therapy (SMART)) was a generous gift from the International Rett Syndrome Foundation.

**Plasmids.** The pCMV6-Arc-Myc-DDK (FLAG) plasmid (MR206218) was purchased from OriGene Biotechnologies (Rockville, MD). The Arc-Myc-FLAG K24Q, K24R, K33Q, K55Q, K70Q, and K70R plasmids were generated with the help of the Q5 Site-Directed Mutagenesis Kit from New England BioLabs (Ipswich, MA) using WT pCMV6-Arc-Myc-FLAG as template. All constructs were verified by DNA sequencing.

**Real-time reverse transcriptase PCR.** After experimental treatment, total RNA was isolated from primary cortical neuron cultures using the TRIzol method (Invitrogen). The concentration of total RNA was measured using a NanoDrop 1000 spectrophotometer (Thermo Fisher Scientific, Rockford, IL) and first-strand complementary DNA (cDNA) was synthesized using the iScript cDNA Synthesis Kit (Bio-Rad, Hercules, CA). Real-time PCRs were performed using gene-specific primers and monitored by quantification of SYBR Green I fluorescence using a Bio-Rad CFX96 Real-Time Detection System. Expression was normalized against Gapdh expression. The relative quantification from at least three biological replicates was performed using the comparative cycle threshold ($\Delta\Delta C_T$) method.

Primers for real-time reverse transcription PCR experiments were: Arc, 5′-TAGCCAGTGACAGGACCCAG-3′ (forward) and 5′-CAGCTCAAGTCC TAGTTGGCAAA-3′ (reverse); Map2, 5′-GGATTTCCATACAGAGAGG AGGAG-3′ (forward) and 5′-CCGTTGATCCCGTTCTCTTTG-3′ (reverse); GluA1, 5′-AGCCAGAATCATGCAGCAGTG-3′ (forward) and 5′-CGCCATCA CCTTCACACCATC-3′ (reverse); GluA2, 5′-TTCGGCCCTGACTTATGATGC-3′ (forward) and 5′-GGCCAAACAATCTCCTGCATTTC-3′ (reverse); Gapdh, 5′-ATGACCACAGTCCATGCCATC-3′ (forward) and 5′-CCAGTGGATGCAGG GATGATGTTC-3′ (reverse).

**Western blotting.** For western blot analyses, cells were collected by scraping in ice-cold radioimmunoprecipitation assay (RIPA) buffer (50 mM Tris-HCl (pH 8.0),

300 mM NaCl, 0.5% Igepal-630, 0.5% deoxycholic acid, 0.1% SDS, 1 mM EDTA) supplemented with a cocktail of protease inhibitors (Complete Protease Inhibitor without EDTA, Roche Applied Science, Indianapolis, IN) and phosphatase inhibitors (phosphatase inhibitor cocktail A, Santa Cruz Biotechnology). One volume of 2× Laemmli buffer (100 mM Tris-HCl (pH 6.8), 4% SDS, 0.15% bromophenol blue, 20% glycerol, 200 mM β-mercaptoethanol) was added and the extracts were boiled for 5 min. Samples were adjusted to an equal concentration after protein concentrations were determined using the BCA assay (Pierce, Thermo Fisher Scientific). Lysates were separated using SDS–PAGE and transferred to a nitrocellulose membrane. After transfer, the membrane was blocked in TBST (Tris-buffered saline + 0.1% Tween 20) supplemented with 5% nonfat powdered milk and probed with the indicated primary antibody at 4 °C overnight. After washing with TBST, the membrane was incubated with the appropriate secondary antibody and visualized using enhanced chemiluminescence (ECL) reagents according to the manufacturer's guidelines (Pierce, Thermo Fisher Scientific). Uncropped scans with molecular weight reference are shown in Supplementary Figs. 10–20.

The following procedure was used to quantify western blots. First, equal quantity of protein lysate was analyzed by SDS–PAGE for each biological replicate. Second, the exposure time of the film to the ECL chemiluminescence was the same for each biological replicate. Third, all the exposed films were scanned on a Brother MFC-J485DW scanner in grayscale at a resolution of 300 dpi. Fourth, the look-up table (LUT) of the scanned tiff images was inverted and the intensity of each band was individually measured using the selection tool and the histogram function in Adobe Photoshop CC 2015 software. Finally, the intensity of each band was divided by the intensity of its respective loading control (β-actin) to provide the normalized value used for statistical analysis.

**Behavior and tissue preparation**. Male mice (C57BL/6 strain) were received as 6-weeks old and habituated to their home cage as trios for 2 weeks before conducting the experiments. During the habituation time period, mice were kept on a light–dark cycle with minimal disruption and unrestricted access to food and water. On experiment day, mice from the same cage were taken and randomly subjected to intraperitoneal injection of vehicle (25% Kolliphor EL (Sigma-Aldrich, C5135), 10% DMSO) or AK-7 (30 mg kg$^{-1}$) solution and immediately placed back in their home cage or transferred to a novel environment. The novel environment consisted of a large cage placed in a different room with new bedding, food, as well as multiple objects like tunnel, running wheel, igloo, and aspen labyrinth that the mice could freely explore. About 3 h after the injection procedure, mice were rapidly anesthetized and killed by decapitation. Finally, brain tissues were harvested and flash frozen in a dry ice/isopentane bath and stored at −80 °C. All experiments were approved by the Animal Care and Use Committee of the Massachusetts General Hospital and carried out according to institutional guidelines.

**Immunocytochemistry**. Indirect immunofluorescence detection of antigens was carried out using cortical neurons cultured on poly-L-lysine/laminin-coated coverslips in 24-well plates at a density of $0.1 \times 10^7$ per well. After experimental treatment, cells were washed twice with PBS and fixed for 30 min at room temperature with 4% paraformaldehyde in PBS. After fixation, cells were washed twice with PBS, permeabilized with PBST (PBS + 0.25% Triton X-100) for 20 min, blocked in blocking solution (PBS + 5% normal goat serum) for another 30 min, and finally incubated overnight at 4 °C with the first primary antibody in blocking solution. The next day, coverslips were extensively washed with PBS and incubated for 2 h at room temperature in the appropriate fluorophore-conjugated secondary antibody solution (Alexa 488- or Alexa 594-conjugated secondary antibody (Molecular Probes, Invitrogen) in blocking solution). After washing with PBS, the coverslips were incubated again overnight in primary antibody solution for the second antigen, and the procedure for conjugation of the fluorophore-conjugated secondary antibody was repeated as above. Finally, cell nuclei were counterstained with 4′,6-diamidino-2-phenylindole (DAPI), and coverslips were mounted on glass slides with ProLong Antifade reagent (Invitrogen).

Cells cultured on coverslips from three independent biological replicates were imaged with an IN Cell Analyzer 6000 high-performance laser-based confocal imaging system and a ×40 objective (GE Healthcare Life Sciences, Marlborough, MA). Image preparation, assembly, and analysis were performed with ImageJ (NIH) and Photoshop CS. Change in contrast and evenness of the illumination was applied equally to all images presented in the study. The following procedure was used for low-throughput measurements. First, original raw tiff files were opened in ImageJ, the nucleus of all neurons in the image was located based on Map2 immunostaining, then average pixel intensity corresponding to Arc immunofluorescence was measured for a 30-pixel spot positioned at the center of the nuclear compartment. Second, for each measure of Arc nuclear immunofluorescence pixel intensity, a measure of background pixel intensity from the same image channel was acquired and subsequently subtracted from the Arc nuclear immunofluorescence pixel intensity value. Finally, Arc immunofluorescence signal from untreated samples was used to establish an objective threshold (two standard deviations above the nuclear Arc immunofluorescence signal averaged from a representative population of untreated neurons) and allow comparison of nuclear Arc expression between different experimental conditions.

**Chemogenomic screen**. Screening of compounds in the chemogenomic library was carried out using DIV13 primary forebrain cortical neurons cultured on poly-L-lysine/laminin-coated 96-well plates (Ibidi USA, Madison, WI). Cells were seeded at a density of 30,000 cells per well and treated with the compound library and BDNF according to the experimental design presented in Fig. 3a. After experimental treatment, cells were processed for Arc and acetyl-H3K9 immunocytochemistry according to the protocol described above. Both antigens (Arc and acetyl-H3K9) were imaged using an IN Cell Analyzer 6000 imaging system and a ×40 objective. Sufficient number of high-resolution images was collected from three independent biological replicates to measure nuclear Arc and acetyl-H3K9 immunofluorescence signals in at least 300 cells for each tested compounds. High-throughput quantification of nuclear Arc immunofluorescence signal was performed in a similar fashion than the low-throughput procedure described above but with the exception that acetyl-H3K9 was used as a nuclear mask to guide the automated quantification of nuclear Arc immunofluorescence signal.

**Biochemical HDAC assay**. BPS Bioscience (San Diego, CA) services were used to estimate the inhibitory activities of AK-7 and CI-994 toward class I/IIa/IIb HDACs using assays of the deacetylase activity of recombinant forms of each HDAC.

**Immunohistochemistry**. Coronal sections containing the hippocampus from all animals were cut from the frozen blocks at a thickness of 20 μm with a CM3050 cryostat (Leica, Nussloch, Germany). The cut sections were captured on glass slides subbed with Vectabond reagent (Vector Laboratories, Burlingame, CA), air dried, and then maintained at −80 °C until immunohistological processing.

Single immunohistochemical staining was used to detect the presence of Arc or acetyl-H3K9 antigen. Sections were fixed for 10 min in 4% paraformaldehyde solution, followed by a 5 min PBS wash. Slides were then incubated for 30 min in 5% normal goat serum in PBS, followed by overnight incubation with mild agitation at 4 °C in primary antibody solution (PBS with anti-Arc or anti-acetyl-H3K9 antibody and 5% normal goat serum). Sections were then washed three times in PBS, followed by incubation for 2 h at room temperature in Alexa-488-conjugated secondary antiserum (1:500 dilution in PBS and 5% normal goat serum). After incubation, sections were counterstained with DAPI and immediately coverslipped with ProLong AntiFade mounting medium (Molecular Probes, Invitrogen).

The following approach was adopted to quantify the number of Arc-positive cell bodies and average acetyl-H3K9 immunostaining pixel intensity in dentate gyrus. First, complete dentate gyrus overlapping grayscale images of DAPI fluorescence and Arc or acetyl-H3K9 immunostaining were collected using a ×20 objective and Nikon Eclipse 90i upright microscope equipped with a motorized stage and image stitching capability. Second, the boundary of the dentate gyrus based on the DAPI fluorescence was determined in each image set with the ImageJ wand tool. Finally, this information was used for image segmentation and quantification of Arc-positive cell bodies or average acetyl-H3K9 immunostaining pixel intensity within each dentate gyrus images. These two tasks were performed with the ImageJ particle analysis and measure tool, respectively. Data for number of Arc-positive cell bodies was normalized and presented according to dentate gyrus area (mm$^2$).

**Detection of polyubiquitinated Arc**. To assess polyubiquitination of Arc in DIV13 primary cortical neurons, Arc was immunopurified from total lysates using an agarose-conjugated anti-Arc monoclonal antibody (Santa Cruz Biotechnology, clone C-7). Overexpressed WT and mutant Arc-Myc-FLAG in Neuro2a cells were immunopurified from total lysates with anti-FLAG M2 antibody conjugated to magnetic beads (Sigma-Aldrich, M8823). Immunopurified materials were separated using SDS–PAGE and electrophoretically transferred to a nitrocellulose membrane. To enhance the immunodetection of ubiquitin conjugated to Arc, the nitrocellulose membrane was placed between two layers of Whatman filter paper, submerged in distilled water, and autoclaved for 30 min. After autoclaving, the membrane was blocked in blocking solution (TBST supplemented with 5% milk) for 30 min and incubated overnight with an antibody recognizing ubiquitin (Santa Cruz Biotechnology, clone P4D1, 1:200) in blocking solution. Subsequent steps were carried out according to the western blot procedure described above.

**Detection of acetylated Arc**. To assess Arc lysine acetylation in DIV13 primary cortical neurons and Neuro2a cells overexpressing WT Arc-Myc-FLAG, the immunopurification procedure was similar to that described above for the detection of polyubiquinated Arc. Immunopurified material was processed according to the western blot methodology from above and the membrane probed with the acetylated lysine (Ac-K$^2$-100) antibody (HRP conjugate) from Cell Signaling Technology.

**Mass spectrometric analysis**. The experimental procedure used to isolate sufficient amount of Arc protein for detection of acetylated and ubiquitinated lysine residues by mass spectrometry is summarized in Supplementary Fig. 9. Briefly, Arc-Myc-FLAG overexpressed in Neuro2a cells was immunopurified with anti-FLAG M2 antibody conjugated to magnetic beads. After washes with RIPA buffer, Arc-Myc-FLAG was eluted from the FLAG antibody by incubating the beads in 50

μl of RIPA buffer containing 25 μg of FLAG peptide (Sigma-Aldrich, F3290) for 2 h at 25 °C with gentle agitation.

Eluates from nine separate IPs were pooled together, concentrated by ethanol protein precipitation, and separated by SDS–PAGE. After Coomassie staining, the gel band corresponding to Arc-Myc-FLAG was excised and in-gel digested using trypsin prior to mass spectrometric analysis. All LC/MS experiments were performed by using a Q Exactive mass spectrometer (Thermo Scientific) coupled to a micro-autosampler AS2 and a nanoflow HPLC pump (Eksigent Technologies, Dublin, CA). Peptides were separated using an in-house packed C18 analytical column (Magic C18, 3 μm particle size, 200 Å pore size (Michrom BioResources, Auburn, CA)) by a 60 min linear gradient starting from 95% Buffer A (0.1% (v/v) formic acid in HPLC water) and 5% Buffer B (0.1% (v/v) formic acid in acetonitrile) to 35% Buffer B. Loading buffer was Buffer A plus 5% formic acid. A full mass spectrum with resolution of 70,000 (relative to an $m/z$ of 200) was acquired in a mass range of 300–1500 $m/z$ (AGC target $3 \times 10^6$, maximum injection time 20 ms). The 10 most intense ions were selected for fragmentation via higher-energy c-trap dissociation (HCD; resolution 17,500, AGC target $2 \times 10^5$, maximum injection time 250 ms, isolation window 1.6 $m/z$, normalized collision energy 27%). MS raw data were analyzed using ProteinPilot (version 4.5.1; Paragon Algorithm 4.5.1, SCIEX) using the mouse UniProtKB database (version 02-2012). The "Thorough ID" search mode was applied with the special factors set to "phosphorylation emphasis" and "gel-based ID". A confidence score of 99 was required for a peptide for the Paragon Algorithm. All modification sites were manually confirmed by interrogating the data.

**Surface AMPAR imaging**. Our procedure to label surface AMPARs containing the GluA1 subunit was adapted from two reports[14,61]. Precisely, DIV14 primary cortical neurons cultured on coverslips were rinsed once with room temperature PBS$^{MC}$ (PBS containing 0.5 mM $CaCl_2$, 1 mM $MgCl_2$, and 4% sucrose). Live cells were then incubated for 30 min at room temperature with GluA1 primary antibody solution (PBS$^{MC}$ with antibody against the extracellular N-terminus of GluA1 (EMD Millipore Corp, clone RH95, 5.0 μg ml$^{-1}$)). Next, cells were rinsed twice with ice-cold PBS$^{MC}$ followed by fixation with 4% paraformaldehyde + 4% sucrose solution for 15 min at room temperature. After fixation, neurons were washed with regular PBS and then incubated overnight with Map2 primary antibody solution (PBS with 0.125% Triton X-100, 5% normal goat serum, and anti-Map2 antibody) to label dendrites. Next day, coverslips were washed with PBS, incubated with appropriate secondary antibodies for 1 h at room temperature, and finally mounted on glass slides with ProLong Antifade reagent (Invitrogen).

Quantification of surface GluA1 puncta was performed according to the following strategy. First, overlapping digital images of GluA1 and Map2 immunostainings were collected using a Zeiss AxioObserver Z1 inverted microscope and ×63 oil immersion objective. Second, thresholding and segmentation based on the Map2 images was used to remove GluA1 signals outside of dendritic processes. Importantly, the same background/foreground cutoff threshold was used for images of all experimental conditions. Third, the length of dendrites for each image set was measured and recorded. Finally, distinct GluA1 foreground objects along dendrites >5 pixels were counted using the particle analysis tool in ImageJ and presented as the average number of GluA1 puncta per μm of dendrite length for each experimental condition. A similar procedure was used for quantification of Arc puncta along dendrites shown in Fig. 4d.

**Electrophysiology**. Primary cortical neurons cultured on glass coverslips were recorded at DIV15 in voltage clamp mode. The external solution was kept at 25 °C and contained 120 mM NaCl, 2 mM KCl, 1 mM $MgCl_2$, 1.5 mM $CaCl_2$, 10 mM 4-(2-hydroxyethyl)-1-piperazineethane-sulfonic acid (HEPES) and 15 mM glucose. The external solution (pH 7.35) supplemented with AP5 (100 μM), tetrodotoxin (1 μM), and bicuculline (50 μM) to isolate AMPAR currents. Patch recording pipettes (3–6 MΩ) were filled with an internal solution (pH 7.2). The internal solution contained 120 mM K-gluconate, 6 mM NaCl, 1 mM $MgCl_2$, 10 mM HEPES, and 0.2 mM ethylene glycol tetraacetic acid. Data were acquired using an HEKA EPC-10 patch-clamp amplifier interfaced with PatchMaster software and analyzed using FitMaster software (HEKA Elektronik, Holliston, MA) and MiniAnalysis (Synaptosoft, Fort Lee, NJ). Miniature EPSCs were recorded for a minimum of 2 min at a holding potential of −80 mV. Analysis of synaptic event data was performed using MiniAnalysis software with event detection threshold set at maximum root mean squared noise level (5 pA). Recordings of synaptic events were assessed for frequency (Hz) and mean amplitude (pA).

**Statistics**. Unless mentioned otherwise, all results represent the mean ± SEM from at least three independent experiments. One-way ANOVA followed by Tukey's post hoc test for multiple comparisons were performed where indicated.

**Data availability**. All data are available within the article and its Supplementary Files, or available from the authors upon request.

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

## Acknowledgements

This work was supported by NIH/NIMH grant MH095088, funding from the Stanley Center for Psychiatric Research, and the Stuart & Suzanne Steele MGH Research Scholars Program to S.J.H. The content is solely the responsibility of the authors and does not necessarily represent the official views of Harvard University, and its affiliated academic healthcare centers, or the NIH. B.A. was supported by the Scientific and Technological Council of Turkey (2214/A International Research Fellowship Program). We thank I. Gaisina (College of Pharmacy, University of Illinois at Chicago) for the generous gift of the SMART compound library through support of the International Rett Syndrome Foundation (IRSF), J. Camacho and the Center for Comparative Medicine at MGH for helping with the novel environment experiments, as well as N. Da Silva (Center for Systems Biology) and the Microscopy Core of the Program in Membrane Biology (PMB) at MGH for providing assistance with microscopy analyses.

## Author contributions

J.L. and S.J.H. designed the research; J.L., S.A.R., S.S., C.S.H., H.W., J.F.S. and W.-N.Z. performed experiments; all authors analyzed the data; J.L. wrote the manuscript with revisions from all other authors.

## Additional information

**Competing interests:** The authors declare no competing financial interests.

