## [Peer Review File · Nature Communications]

Reviewers' comments:

Reviewer #1 (Remarks to the Author):

The study aimed to identify compounds that modulate BDNF-induced Arc expression as potential therapeutics. Using a chemogenomic screen a range of compounds resulting in enhanced or reduced levels of nuclear Arc protein in cultured cortical neuronal was identified. Focusing on the effects of three lysine deacetylase inhibitors (AK-7, oxamflatin, and CI-944) the authors provide evidence that these compounds increase abundance of nuclear Arc protein by inhibiting protein degradation. In a series of experiments using Neuro2a cells overexpressing tagged-Arc, the authors identified sites of lysine acetylation and ubiquitination by mass spectroscopy, and further showed that K to Q substitutions on two of the modified lysine residues (K24 and K70) reduced levels of polyubiquitinated Arc and slowed the rate of Arc protein degradation. Finally, in mice exploring a novel environment, intraperitoneal injection of AK-7 increased the intensity of Arc protein expression in granule cells of the dentate gyrus.

In sum, the findings on Arc acetylation are novel, but require further substantiation. More importantly in this context, the mechanism and function of Arc acetylation are unknown.

1. The analysis in Neuro2A cell lines was based entirely on overexpression of pseudo-acetylated (K/Q) mutants. The rigor of this basic analysis would be improved by comparison with a pseudo-de-acetylated (K/R) mutant.
2. It is important to establish the function of Arc acetylation, for instance by expression acetylation mutants in primary neuronal cultures. How does acetylation of specific lysine residues impact homeostatic plasticity, AMPA receptor trafficking, and spine morphology?
3. Imaging of Arc in the nucleus was convenient in the initial high-throughput screening of compounds. In the subsequent analysis, it is important to assess the role of acetylation on the subcellular localization of Arc in neuronal dendrites and spines, comparing K/Q with K/R.
4. The systemic injection of AK-7 could have effects on numerous brain circuits regulating dentate gyrus activity and impact numerous biochemical pathways in dentate granule cells affecting Arc protein turnover.
5. Figure 1 confirms BDNF-induced increases in Arc mRNA and protein expression and

provides pharmacological evidence for involvement of Rac signaling in Arc protein expression. However, the analysis of transcription/translation regulation is preliminary and disconnected from the rest of the study. In Fig 4 the lack of effect of KDAC inhibitors on ERK and rp6 is only of tangential interest, given that mRNA levels are not changed, and effects on protein half-life are determined in the CHX experiments.

6. The acetyltransferases and deacetylases involved are unknown.

Minor:

Arc antibody C7 should be indicated in Materials and Methods.

Pg. 15. "Fig 3" should be Fig. 8

Reviewer #2 (Remarks to the Author):

The manuscript titled "Chemogenomic Analysis Reveals Key Role for Lysine Acetylation in Regulating Arc Stability" by the authors Lalonde et al., is a well written manuscript detailing the SMART compound screening in BDNF mediated nuclear enrichment or depletion of Arc protein.

1. Their results are novel in the identification of AK7 or similar compounds influencing non-histone lysine acetylation of Arc at K24 and K70 respectively. Further, they also identify novel lysine acetylation and ubiquitination sites on Arc namely K92, K136, K33 and K56 using MS/MS analysis.
2. Using a battery of in vitro experiments the authors demonstrate increase in nuclear Arc in the presence of BDNF when cultures are pretreated with AK7 and Oxamflatin. The authors further confirm the results in vivo after stereotactic injection of AK7 into DG granule cells. In the end of the manuscript one is left wondering what is the consequence of Arc stabilization in neuronal nucleus via K24 and K70 acetylation. This remains one of the main draw back of this manuscript.
3. Neuronal Arc regulation and its role in synapse plasticity remains is of great general interest to the neuroscience community. However, authors fail to make use of their stable Arc mutants, namely K24 or K70A in addressing this issue.

4. Lysine modification(s) are prevalent within structured protein domains, often facilitating protein-protein interactions and macro molecular complex assembly. The authors briefly mention the possibility of Tip60 interaction within nucleus after Arc acetylation, but do not provide any experimental evidence supporting this line of thinking.

5. PML bodies have generated significant interest in the nuclear field given their role in recruiting SUMOylated proteins. Arc associates with PML; however, we fail to understand how Arc stabilization influences PML distribution or PML body composition.

6. In relation to the above point, Arc was shown to be a novel SUMO substrate (Craig et al., 2012) with SUMOylation at K110 and K268. While authors claim a novel interaction between Acetylation and Ubiquitination pathways, a recent report demonstrated a cross talk between Acetylation, phosphorylation and SUMOylation of the neuronal protein gephyrin (Ghosh et al., 2016). Hence, is Arc acetylation also having an influence on its SUMO conjugation?

7. Ubiquitination is well documented in literature as a promiscuous protein modification. If Arc cannot be Ub at K24 and K70 would that influence ubiquitination at K136 also?

8. Overall the study is very interesting and offers fresh insights into Arc protein stability under the influence of BDNF signaling; however, it would significantly enhance the scope of the study if the authors can link the Arc stability changes to synapse plasticity in vitro and/or in vivo. For example, what happens to the mossy fiber sprouting/synapse as a consequence of Arc stability and nuclear enrichment?

The authors should address pending questions that will provide a functional relevance for Arc stabilization within nucleus.

Reviewer #3 (Remarks to the Author):

Summary: In the article “Chemogenomic Analysis Reveals Key Role for Lysine Acetylation in Regulating Arc Stability”, Lalonde et. al. identify a set of small molecules that abrogate ubiquitin-mediated degradation of Arc. Further, they describe a mechanism where HDAC inhibition by the identified small molecules causes acetylation to accumulate on Arc lysines, which block the ability of these same lysines to be ubiquitinated. The authors rigorous experiments strongly support their conclusions. While there are certainly further questions to explore stemming from this work, sharing the present communication with the neurobiology field will stimulate the investigations of these new avenues. The manuscript is well written and I suggest only minor textual edits prior to publication in Nature Communications.

Minor Comments:

1) In Figure 5D, I believe the MS/MS spectrum shows diglycine (G-G) as the lysine modification within the digested peptide sequence, not ubiquitin, as the G-G remnant remains after digestion of a ubiquitin-modified protein with trypsin (consistent with the workflow illustrated in Figure S7). The results perfectly support the claim that this is a ubiquitination site, but his subtle distinction should be explained in the figure legend and results section for clarity.

2) The authors should clarify their comments of on Page 13, where they describe key proteomics results by stating that they detected peptides "... containing either an acetylation or ubiquitination mark at lysine residues 24, 33, and 55". It is my interpretation that these three lysines were detected as sites of both acetylation and ubiquinylation, with each PTM detected in separate spectra. As it is currently stated, however, a reader could interpret it as some of these sites contained only one of the PTMs, but not both. This is another subtle distinction, but it is important for the mechanism suggested by the authors, where the sites are susceptible to both PTMs and acetylation can block ubiquitination of the same residues.

3) I suggest some discussion on speculated mechanisms for Arc acetylation and ubiquitination. Previous work has implicated Triad3A in the ubiquitination of Arc (Neuron. 2014 Jun 18;82(6):1299-316.) It has also been previously shown that Arc interacts with the E3 ligase E6AP/UBE3A, but whether it is ubiquitinated by this enzyme has been questioned (Proc Natl Acad Sci U S A. 2013 May 28;110(22):8888-93.). How does the current study fit into this body of literature? Similarly, could Arc's known interaction with the acetyltransferase TIP60 (eNeuro. 2014 Nov 12;1(1)) help explain the mechanism of its acetylation? Even if E6AP/UBE3A and TIP60 do not catalyze these PTMs on Arc, could their interactions suggest that Arc may have other interactors from these same classes of PTM-mediating enzymes? A bit of discussion here would help place the current findings into this previous body of literature.

4) A number of grammatical errors exist throughout the manuscript, which are quite minor but distracting nonetheless. Here are some examples fixes:

- a) Page 2: "... and possibly most versatile, players...";
- b) Page 2: "...to the endocytosis of 3-...";
- c) Page 3: "...contexts could help alleviate cognitive...";
- d) Page 10: "...not only sequestered Arc in the...";
- e) Page 11: "...the same time as CHX and...";
- f) Page 12: "...Arc protein remained elusive...";
- g) Page 14: "...Arc by competing with the ubiquitination..."

Reviewer #4 (Remarks to the Author):

Lalonde et al manuscript describing the role of lysine acetylation in regulating stability of Arc protein in neurons is beautifully written, all the experiments are done well for the most part, used solid statistical analyses to interpret data. There are no major technical concerns except for the overemphasis on the possibility of reversing Arc abnormalities to improve neurological impairments in different disease contexts. The significance of Arc as a therapeutic target needs to be substantiated with experimental evidence.

The current manuscript indeed provides new insight into the biology of Arc protein in neurons. Title is very much appropriate for describing these findings. However the abstract, introduction and discussion sections of the manuscript emphasize on Arc as a therapeutic target.

Along these lines, data is not convincing that Arc is the direct target of the lead compounds discovered by the chemogenomic screen. While this is a problem in general with phenotypic screens, additional experiments are required to demonstrate some sort of target engagement for the lead compounds.

What are the phenotypes of Arc knockout and Arc overexpression in cultures? Could these compounds rescue the phenotype?

Why was Neuro2A cells used for the experiment because the team is able to manipulate expression of Arc in neurons?

Does the AK7 compound demonstrate any memory enhancing properties in their behavioral experiments?

We are grateful for the Reviewers careful and in depth reading of the manuscript and overall constructive feedback. We have now had the chance to conduct a series of additional experiments to directly address the questions raised and, in doing so, have provide greater functional insight into the role that reversible Arc acetylation plays in regulating its degradation through the proteasomal system and synaptic plasticity. Overall, these additional data have significantly strengthened our manuscript and we are pleased to be able to submit this revised version for consideration. Below we provide a point-by-point response in bold to each of the issues raised and have highlighted changes to the manuscript.

REVIEWER #1

The study aimed to identify compounds that modulate BDNF-induced Arc expression as potential therapeutics. Using a chemogenomic screen a range of compounds resulting in enhanced or reduced levels of nuclear Arc protein in cultured cortical neuronal was identified. Focusing on the effects of three lysine deacetylase inhibitors (AK-7, oxamflatin, and CI-944) the authors provide evidence that these compounds increase abundance of nuclear Arc protein by inhibiting protein degradation. In a series of experiments using Neuro2a cells overexpressing tagged-Arc, the authors identified sites of lysine acetylation and ubiquitination by mass spectroscopy, and further showed that K to Q substitutions on two of the modified lysine residues (K24 and K70) reduced levels of polyubiquinated Arc and slowed the rate of Arc protein degradation. Finally, in mice exploring a novel environment, intraperitoneal injection of AK-7 increased the intensity of Arc protein expression in granule cells of the dentate gyrus. In sum, the findings on Arc acetylation are novel, but require further substantiation. More importantly in this context, the mechanism and function of Arc acetylation are unknown.

We are pleased that Reviewer #1 recognizes that our “findings on Arc acetylation are novel”. To address the Reviewer’s concern that our discoveries “require further substantiation”, as outlined below with regards to the specific points raised, we now provide in our revised manuscript a considerable amount of new biochemical, imaging, and electrophysiology data which further support the idea that lysine acetylation impacts the control and function of Arc protein in neurons. A response to the specific issues raised is provided below.

Specific issues:

1. The analysis in Neuro2A cell lines was based entirely on overexpression of pseudo-acetylated (K/Q) mutants. The rigor of this basic analysis would be improved by comparison with a pseudo-de-acetylated (K/R) mutant.

As suggested by the Reviewer, we have now extended our analysis in Neuro2A cells for lysine residues 24 and 70 to pseudo-deacetylated (K/R) mutant constructs (now shown in Supplementary Figure 5). In the first version of our manuscript, analysis with a pseudo-acetylated K/Q mutant of these sites had demonstrated an effect on Arc protein by prolonging its half-life. As expected, the K/R mutant Arc protein of each lysine residue was similarly resistant to degradation since the modification of K to R disrupts the potential for

ubiquitination as well as acetylation. As we specifically mention in the revised manuscript, this result provides support to the idea that a direct competition between the acetylation and ubiquitination systems for a specific site (e.g. K24 for which we provide MS/MS evidence for both PTMs) is sufficient to alter Arc protein abundance.

2. It is important to establish the function of Arc acetylation, for instance by expression acetylation mutants in primary neuronal cultures. How does acetylation of specific lysine residues impact homeostatic plasticity, AMPA receptor trafficking, and spine morphology?

We completely agree with this comment on the importance, as the Reviewer is suggesting, of establishing how the acetylation of specific Arc lysine may impact synaptic plasticity. To better address these questions, in our revised manuscript we now include a description and data (see below) from a series of additional experiments designed to precisely address these questions summarized as follows.

First, as now shown in revised Figure 4C and D (see below) we conducted a comparative analysis of Arc puncta distribution along dendrites of primary cortical neurons. The goal of this experiment was to specifically assess how the three KDAC inhibitors found with our screen are influencing the fraction of Arc protein that is locally synthesized in proximity to postsynaptic terminals. Here, our results clearly show that treatment with these KDAC inhibitors not only increase level of nuclear Arc (screening data, Figure 3B and C), but also enhance the number of Arc puncta visible along dendrites providing an example of the role of modulating KDAC activity in determining Arc’s localization into multiple subcellular compartments.

Figure 4. Post-screen validation of KDAC inhibitors AK-7, oxamflatin, and CI-994 potentiating effect on Arc protein abundance. (A) Western blot analysis reveals that lysates from BDNF-treated cortical cultures supplemented with AK-7, oxamflatin, or CI-994 have significantly more total Arc protein than BDNF plus vehicle (DMSO) control condition. Chemical structure of each compound is presented on top. (B) Graphs show mean ($n = 5$) Arc/ β -actin ratio (\pm SEM) for cells treated as in (A). One-way ANOVA revealed a significant dose-response difference between compound concentrations [AK-7, $F_{2,12} = 4.48$, $p < 0.05$; oxamflatin, $F_{2,12} = 46.72$, $p < 0.0001$; CI-994, $F_{2,12} = 12.24$, $p < 0.005$]. Tukey’s HSD post hoc test, * $p < 0.05$; ** $p < 0.01$, *** $p < 0.005$; **** $p < 0.0001$. Western blots run with the same sample lysates revealed that these three KDAC inhibitors do not alter BDNF-dependent phosphorylation of p44/42-Mapk and rpS6. (C) Each tested KDAC inhibitor increases number of Arc puncta along dendrites labeled by Map2 immunocytochemistry. Representative captures of DIV14 mouse primary cortical neurons from each experimental condition co-immunostained for Arc (red fluorophore) and Map2 (green fluorophore). White arrowheads show examples of discrete Arc puncta. Primary cortical neurons were treated with BDNF and AK-7, oxamflatin, or CI-994 at a final concentration of 16.7 μ M. (D) Graph shows mean Arc puncta per dendrite μ m for cells treated as in (C). Separate ANOVA revealed a significant difference between cells co-treated with a KDAC and BDNF over untreated and BDNF alone conditions [AK-7, $F_{2,82} = 12.82$, $p < 0.0001$; oxamflatin, $F_{2,78} = 46.31$, $p < 0.0001$; CI-994, $F_{2,82} = 8.08$, $p < 0.001$]. Tukey’s HSD post hoc test, * $p < 0.05$; *** $p < 0.005$; **** $p < 0.0001$.

Second, as now shown in revised Figure 8A and B (see below), we performed additional experiments that characterized the abundance of surface AMPARs containing the GluA1 subunit along dendrites upon treatment with our KDAC inhibitors. Since expression of dendritic/synaptic Arc is well known to favor endocytosis of AMPARs (1-3), we reasoned that the increased abundance of Arc puncta caused by application of KDAC inhibitors, as shown in revised Figure 4C and D, should impact levels of surface AMPAR in a consistent manner. Excitingly, as predicted, treatment of primary cortical neurons with BDNF and with either one of the tested KDAC inhibitor caused a significant decrease in surface GluA1 as compared to control conditions (untreated and BDNF alone conditions). Importantly, these results are concordant with those collected in Figure 4C and D. Furthermore, to rule out that this change may be attributable to a reduction in *GluA1* mRNA levels, we performed a quantitative real-time PCR analysis and found no change in expression between key experimental conditions (Supplementary Figure 6).

Figure 8. Effects of KDAC inhibitors AK-7, oxamflatin, and CI-994 on AMPAR biology. (A) Representative images of surface GluA1 immunostaining in DIV14 mouse primary cortical neurons for each tested experimental condition. AK-7, oxamflatin, or CI-994 was applied at a final concentration of 16.7 μM and treatment duration was 6 h with BDNF. (B) Graph shows mean number of surface GluA1 puncta per dendrite μm for cells treated as in (A). Separate ANOVA revealed a significant difference between cells co-treated with a KDAC inhibitor and BDNF over untreated and BDNF alone conditions [AK-7, $F_{2,72} = 7.28$, $p < 0.005$; oxamflatin, $F_{2,72} = 10.37$, $p < 0.0005$; CI-994, $F_{2,72} = 4.25$, $p < 0.05$]. Tukey's HSD post hoc test, * $p < 0.05$; ** $p < 0.01$; *** $p < 0.005$. (C) Representative traces of mEPSCs from DIV15 mouse primary cortical neurons treated according to the indicated treatment (untreated, BDNF alone, or AK-7 plus BDNF). Cells that were co-treated with AK-7 and BDNF for 6 h prior to recording show lower mEPSC amplitude. (D) Quantification of mEPSC amplitude for experimental conditions shown in (C). One-way ANOVA revealed a significant difference in mEPSC amplitude between the BDNF alone and AK-7 plus BDNF condition [$F_{2,22} = 3.89$, $p < 0.05$]. Tukey's HSD post hoc test, * $p < 0.05$. A total of 5086 (untreated, $n = 9$ cells), 3760 (BDNF alone, $n = 8$ cells), and 4475 (AK-7 plus BDNF, $n = 8$ cells) mEPSC events were analyzed from three independent cultures. (E) Quantification of mEPSCs frequency shows no differences between experimental conditions.

Third, to gain deeper insight into the effects of modulating Arc on neurophysiology, we extended our biochemical observations regarding the interaction between surface GluA1 levels and treatment with the KDAC inhibitor AK-7 to perform single-cell electrophysiological recordings. As now shown in revised Figure 8C-E, primary cortical neurons co-treated with BDNF and AK-7 displayed a significant reduction in the amplitude, but not the frequency, of mEPSCs in comparison to cells treated with BDNF alone. Taken together, our previous findings and the additional new data presented in Figure 4 and 8 strongly suggest an influence of lysine acetylation on Arc function.

3. *Imaging of Arc in the nucleus was convenient in the initial high-throughput screening of compounds. In the subsequent analysis, it is important to assess the role of acetylation on the subcellular localization of Arc in neuronal dendrites and spines, comparing K/Q with K/R.*

This is an excellent point that the Reviewer is raising. Indeed, for the purpose of developing a robust and scalable high-throughput, image-based screen in our primary cortical neuron culture system, we focused on the quantification of changes in nuclear Arc, which shows strong and dynamic upregulation in response to neuronal stimulation with BDNF and, as we showed, turned out to be highly responsive to diverse pharmacological agents in our library. This strategy led us to our discovery of the profound effects of KDAC inhibitors on Arc protein stability.

In response to the suggestion regarding Arc changes in other subcellular compartments, as described above and now shown in revised Figure 4C and D, we performed a comparative analysis of Arc puncta distribution along dendrites in our primary cortical neuron culture system to specifically assess how the three KDAC inhibitors found with our screen are influencing the fraction of Arc protein that is locally synthesized in proximity to post-synaptic terminals. Here, our results clearly show that treatment with these KDAC inhibitors not only increase levels of nuclear Arc (screening data, Figure 3B and C), but also enhances the number of Arc puncta visible along dendrites providing an example of the role of modulating KDAC activity in determining Arc's localization into multiple subcellular compartments. Future studies with appropriate acetyl-lysine residue specific antibodies will need to be used to perform a more detailed analysis of a particular acetylation site and consideration of the temporal dynamics of changes in Arc subcellular localization.

4. *The systemic injection of AK-7 could have effects on numerous brain circuits regulating dentate gyrus activity and impact numerous biochemical pathways in dentate granule cells affecting Arc protein turnover.*

We acknowledge that, like other experimental manipulations of molecular targets *in vivo* in the brain including genetic approaches, there are limitations in pharmacological studies in particular when compounds are administered systemically. Here, though, to directly address the point of target engagement in the tissue of interest we would like to point out that the acetyl-H3K9 immunofluorescence signal in the dentate gyrus of mice intraperitoneally injected with AK-7 was significantly higher than in the signal measured from animals treated with the vehicle solution (Supplementary Figure 7). This effect of AK-7 on acetyl-H3K9 immunofluorescence matches the observation we made in our screen with primary cortical neurons for the same KDAC inhibitor (Figure 3C). While we cannot rule out that the molecular and cellular mechanisms controlling Arc protein turnover are different in our *ex vivo* primary neuron assays and *in vivo*, this parallel demonstration of a degree of overlap in the molecular impact of AK-7 between the two experimental systems helps establish the correlation between KDAC target engagement and effects on acetylation and changes in levels of Arc. Certainly, we look forward in the future to a better understanding of the enzymatic effectors responsible for Arc acetylation/deacetylation as

this knowledge will then allow the design of more precise and better controlled experiments with animal models. Nonetheless, translating our observations from the *ex vivo* culture system to the *in vivo* context we think makes an important step forward and lays the foundation for conducting exactly such studies in the future.

5. Figure 1 confirms BDNF-induced increases in Arc mRNA and protein expression and provides pharmacological evidence for involvement of Rac signaling in Arc protein expression. However, the analysis of transcription/ translation regulation is preliminary and disconnected from the rest of the study. In Fig 4 the lack of effect of KDAC inhibitors on ERK and rp6 is only of tangential interest, given that mRNA levels are not changed, and effects on protein half-life are determined in the CHX experiments.

Our goal for testing the effect of KDAC inhibitors on ERK (Mapk) and rp6 phosphorylation in Figure 4A was to highlight the importance of systematically considering the different signaling elements involved in BDNF-induced Arc expression were also intact and relevant in our primary cortical neuron culture system that has been adapted for the purpose of high-throughput screening. We can understand that this Reviewer finds this specific data “*of tangential interest*” considering that we also provide quantification of mRNA levels; however, we believe that the systematic approach that we have adopted in our manuscript might be helpful to other researchers interested at understanding how other pharmacological agents found in our screen, other than those ones affecting KDAC activity and Arc acetylation, might act to influence BDNF-induced Arc expression. In particular, these assays of ERK and rp6 phosphorylation may be useful for dissecting the effects of pharmacological agents that suppress Arc expression through affecting transcription or translation rather than the induction of Arc that we focused on.

6. The acetyltransferases and deacetylases involved are unknown.

Having identified the existence of Arc lysine acetylation for the first time and demonstrated a role for its removal by members of lysine deacetylase (KDAC) family in controlling Arc stability in a manner correlated with the effects we shown on AMPARs and electrophysiology, we agree that further investigation of the molecular mechanisms controlling Arc acetylation is warranted. Based upon the known *in vitro* selectivity of the principal KDAC inhibitors we use, namely AK-7, an inhibitor of NAD⁺-dependent Class III lysine deacetylases in the sirtuin family, as well as CI-994 and oxamflatin, which share in common the inhibition of the zinc-dependent, Class I HDACs (HDAC1/2/3), we surmise that multiple KDAC family members may be involved in reversing Arc acetylation and therefor, as we show for the first time, impacting its stability.

In terms of the question of the relevant acetyltransferases, we agree that elucidating their identity is now of great interest. As we performed for the modulation of Arc KDAC activity, such studies will benefit from the discovery in the future of appropriate pharmacological tools for temporal control of Arc acetyltransferase as well as developing appropriate functional genomic tools (RNAi, CRISPR) for control of Arc acetylation. As we mention in the Discussion, there are likely multiple pathways contributing to Arc acetylation, and a systematic analysis of each possibilities will be needed in order to

uncover which acetyltransferases and deacetylases are involved in the control of distinct Arc lysine residues. As completion of these studies are beyond the scope of the current study, we hope that the publication of our findings will serve as a foundation to stimulate precisely these types of studies, and we look forward to future work from our laboratory and others that will shed light on these intriguing questions.

REVIEWER #2

The manuscript titled "Chemogenomic Analysis Reveals Key Role for Lysine Acetylation in Regulating Arc Stability" by the authors Lalonde et al., is a well written manuscript detailing the SMART compound screening in BDNF mediated nuclear enrichment or depletion of Arc protein.

Reviewer #2 found that our study is “overall very interesting and offers fresh insights into Arc protein stability”. The Reviewer provides thoughtful remarks and suggestions.

1. Their results are novel in the identification of AK7 or similar compounds influencing non-histone lysine acetylation of Arc at K24 and K70 respectively. Further, they also identify novel lysine acetylation and ubiquitination sites on Arc namely K92, K136, K33 and K56 using MS/MS analysis.

We appreciate that this Reviewer highlights the novelty of our findings, which through our combined chemogenomic screening approach and secondary mass spectrometry analysis, presents the first evidence for K33, K56, K92, and K136 as candidate acetylation and/or ubiquitination sites.

2. Using a battery of in vitro experiments the authors demonstrate increase in nuclear Arc in the presence of BDNF when cultures are pretreated with AK7 and Oxamflatin. The authors further confirm the results in vivo after stereotactic injection of AK7 into DG granule cells. In the end of the manuscript one is left wondering what is the consequence of Arc stabilization in neuronal nucleus via K24 and K70 acetylation. This remains one of the main draw back of this manuscript.

In the revised version of our manuscript, as outlined in response to Reviewer #1’s similar questions, we now include a description and data (see below) from a series of additional experiments designed to precisely address the question of the consequence of Arc stabilization through acetylation summarized as follows.

First, as now shown in revised Figure 4C and D (see below) we conducted a comparative analysis of Arc puncta distribution along dendrites of primary cortical neurons. The goal of this experiment was to specifically assess how the three KDAC inhibitors found with our screen are influencing the fraction of Arc protein that is locally synthesized in proximity to postsynaptic terminals. Here, our results clearly show that treatment with these KDAC inhibitors not only increase level of nuclear Arc (screening data, Figure 3B and C), but also enhance the number of Arc puncta visible along dendrites providing an example of the role of modulating KDAC activity in determining Arc’s localization into multiple subcellular compartments.

Figure 4. Post-screen validation of KDAC inhibitors AK-7, oxamflatin, and CI-994 potentiating effect on Arc protein abundance. (A) Western blot analysis reveals that lysates from BDNF-treated cortical cultures supplemented with AK-7, oxamflatin, or CI-994 have significantly more total Arc protein than BDNF plus vehicle (DMSO) control condition. Chemical structure of each compound is presented on top. (B) Graphs show mean ($n = 5$) Arc/ β -actin ratio (\pm SEM) for cells treated as in (A). One-way ANOVA revealed a significant dose-response difference between compound concentrations [AK-7, $F_{2,12} = 4.48$, $p < 0.05$; oxamflatin, $F_{2,12} = 46.72$, $p < 0.0001$; CI-994, $F_{2,12} = 12.24$, $p < 0.005$]. Tukey's HSD post hoc test, * $p < 0.05$; ** $p < 0.01$, *** $p < 0.005$; **** $p < 0.0001$. Western blots run with the same sample lysates revealed that these three KDAC inhibitors do not alter BDNF-dependent phosphorylation of p44/42-Mapk and rpS6. (C) Each tested KDAC inhibitor increases number of Arc puncta along dendrites labeled by Map2 immunocytochemistry. Representative captures of DIV14 mouse primary cortical neurons from each experimental condition co-immunostained for Arc (red fluorophore) and Map2 (green fluorophore). White arrowheads show examples of discrete Arc puncta. Primary cortical neurons were treated with BDNF and AK-7, oxamflatin, or CI-994 at a final concentration of 16.7 μ M. (D) Graph shows mean Arc puncta per dendrite μ m for cells treated as in (C). Separate ANOVA revealed a significant difference between cells co-treated with a KDAC and BDNF over untreated and BDNF alone conditions [AK-7, $F_{2,82} = 12.82$, $p < 0.0001$; oxamflatin, $F_{2,78} = 46.31$, $p < 0.0001$; CI-994, $F_{2,82} = 8.08$, $p < 0.001$]. Tukey's HSD post hoc test, * $p < 0.05$; *** $p < 0.005$; **** $p < 0.0001$.

Second, as now shown in revised Figure 8A and B (see below), we performed additional experiments that characterized the abundance of surface AMPARs containing the GluA1 subunit along dendrites upon treatment with our KDAC inhibitors. Since expression of dendritic/synaptic Arc is well known to favor endocytosis of AMPARs (1-3), we reasoned that the increased abundance of Arc puncta caused by application of KDAC inhibitors, as shown in revised Figure 4C and D, should impact levels of surface AMPAR in a consistent manner. Excitingly, as predicted, treatment of primary cortical neurons with BDNF and with either one of the tested KDAC inhibitor caused a significant decrease in surface GluA1 as compared to control conditions (untreated and BDNF alone conditions). Importantly, these results are concordant with those collected in Figure 4C and D. Furthermore, to rule out that this change may be attributable to a reduction in *GluA1* mRNA levels, we performed a quantitative real-time PCR analysis and found no change in expression between key experimental conditions (Supplementary Figure 6).

Third, to gain deeper insight into the effects of modulating Arc on neurophysiology, we extended our biochemical observations regarding the interaction between surface GluA1 levels and treatment with the KDAC inhibitor AK-7 to perform single-cell electrophysiological recordings. As now shown in revised Figure 8C-E, primary cortical neurons co-treated with BDNF and AK-7 displayed a significant reduction in the amplitude, but not the frequency, of mEPSCs in comparison to cells treated with BDNF alone.

Taken together, our previous findings, and the additional new data presented in Figure 4 and 8, strongly suggest an influence of lysine acetylation on Arc function in multiple different subcellular compartments.

Figure 8. Effects of KDAC inhibitors AK-7, oxamflatin, and CI-994 on AMPAR biology. (A) Representative images of surface GluA1 immunostaining in DIV14 mouse primary cortical neurons for each tested experimental condition. AK-7, oxamflatin, or CI-994 was applied at a final concentration of 16.7 μM and treatment duration was 6 h with BDNF. (B) Graph shows mean number of surface GluA1 puncta per dendrite μm for cells treated as in (A). Separate ANOVA revealed a significant difference between cells co-treated with a KDAC inhibitor and BDNF over untreated and BDNF alone conditions [AK-7, $F_{2,72} = 7.28$, $p < 0.005$; oxamflatin, $F_{2,72} = 10.37$, $p < 0.0005$; CI-994, $F_{2,72} = 4.25$, $p < 0.05$]. Tukey's HSD post hoc test, $*p < 0.05$; $**p < 0.01$; $***p < 0.005$. (C) Representative traces of mEPSCs from DIV15 mouse primary cortical neurons treated according to the indicated treatment (untreated, BDNF alone, or AK-7 plus BDNF). Cells that were co-treated with AK-7 and BDNF for 6 h prior to recording show lower mEPSC amplitude. (D) Quantification of mEPSC amplitude for experimental conditions shown in (C). One-way ANOVA revealed a significant difference in mEPSC amplitude between the BDNF alone and AK-7 plus BDNF condition [$F_{2,22} = 3.89$, $p < 0.05$]. Tukey's HSD post hoc test, $*p < 0.05$. A total of 5086 (untreated, $n = 9$ cells), 3760 (BDNF alone, $n = 8$ cells), and 4475 (AK-7 plus BDNF, $n = 8$ cells) mEPSC events were analyzed from three independent cultures. (E) Quantification of mEPSCs frequency shows no differences between experimental conditions.

3. Neuronal Arc regulation and its role in synapse plasticity remains is of great general interest to the neuroscience community. However, authors fail to make use of their stable Arc mutants, namely K24 or K70A in addressing this issue.

We agree with the Reviewer that a direct evaluation of Arc residues K24 and K70 demands closer attention and that the study of these sites may provides novel insights about “Arc regulation and its role in synapse plasticity”. As described above and shown in revised Figure 8, we performed single-cell electrophysiological recordings in our primary neuron cortical cultures and show that co-treatment with BDNF and AK-7 caused a significant reduction in the amplitude, but not the frequency, of mEPSCs in comparison to cells treated with BDNF alone. Beyond these studies, we believe that addressing this question would be better suited within the context of a separate study that will globally investigate the factors contributing to the posttranslational modifications of these specific sites. We certainly hope to rapidly pursue these questions following the opportunity to publish our current manuscript.

4. Lysine modification(s) are prevalent within structured protein domains, often facilitating protein-protein interactions and macro molecular complex assembly. The authors briefly

mention the possibility of Tip60 interaction within nucleus after Arc acetylation, but do not provide any experimental evidence supporting this line of thinking.

The Reviewer is correct that we mentioned in the Discussion of our manuscript Tip60 as a “candidate” acetyltransferase for Arc acetylation based upon the elegant studies of the VanDongen laboratory (eNeuro 0019-14.2014) where a role for the interaction of Arc with Tip60 was shown to impact histone H4K12 acetylation that we cite (4). Overall, although we do not provide experimental evidence in our manuscript that would support Tip60 as a *bona fide* acetyltransferase for Arc our intention in this part of our Discussion was to simply suggest a starting point for future research that would aim to elucidate the molecular machinery responsible for Arc acetylation/deacetylation in neurons. As discussed above in response to Reviewer #1’s questions, we agree that elucidating the identity of the *bona fide* Arc acetyltransferase is now of great interest, and we suggest that such studies will benefit from the discovery in the future of appropriate pharmacological tools for temporal control of Arc acetyltransferase as well as developing appropriate functional genomic tools (RNAi, CRISPR) for control of Arc acetylation. As completion of these studies are beyond the scope of the current study we hope that the publication of our findings will serve as a foundation to stimulate precisely these types of studies, and we look forward to future work from our laboratory and others that will shed light on these intriguing questions.

5. PML bodies have generated significant interest in the nuclear field given their role in recruiting SUMOylated proteins. Arc associates with PML; however, we fail to understand how Arc stabilization influences PML distribution or PML body composition.

The Reviewer raises a very interesting question about if and how changes in nuclear Arc protein stability and abundance conferred by lysine acetylation would affect its interaction with other protein, including PML. Unfortunately, at this point we do not have data that could help address this question. We are aware of the interest towards PLM bodies and how it participates in the recruitment of SUMOylated proteins and we hope that publication of our work would stimulate research on this topic from a new perspective.

6. In relation to the above point, Arc was shown to be a novel SUMO substrate (Craig et al., 2012) with SUMOylation at K110 and K268. While authors claim a novel interaction between Acetylation and Ubiquitination pathways, a recent report demonstrated a cross talk between Acetylation, phosphorylation and SUMOylation of the neuronal protein gephyrin (Ghosh et al., 2016). Hence, is Arc acetylation also having an influence on its SUMO conjugation?

We thank the Reviewer for reminding us about the role of SUMOylation in the regulation of Arc. In the revised manuscript, we now explicitly mention this topic with references in an appropriate section of our Discussion. Whether interplay exists between acetylation, ubiquitination, and SUMOylation for Arc is an intriguing question that we are not able to answer at this point. Our work provides specific lysine residues that will allow systematically investigating this question.

7. Ubiquitination is well documented in literature as a promiscuous protein modification. If Arc

cannot be Ub at K24 and K70 would that influence ubiquitination at K136 also?

This question by the Reviewer perfectly illustrates the intricate interactions that may exist between multiple sites and types of posttranslational modifications for a single protein. To appropriately answer the question formulated by the Reviewer, we believe that an exhaustive series of *in vitro* ubiquitination assay would need to be performed using recombinant WT and mutant Arc protein as substrate. In our opinion, this specific question would be more appropriately addressed within the context of a detailed analysis of a specific site like K24.

8. Overall the study is very interesting and offers fresh insights into Arc protein stability under the influence of BDNF signaling; however, it would significantly enhance the scope of the study if the authors can link the Arc stability changes to synapse plasticity in vitro and/or in vivo. For example, what happens to the mossy fiber sprouting/synapse as a consequence of Arc stability and nuclear enrichment?

The authors should address pending questions that will provide a functional relevance for Arc stabilization within nucleus.

We thank the Reviewer for the encouragement to pursue a more detailed set of studies addressing the question of the link between Arc nuclear levels and effects on synaptic plasticity. As discussed above for Reviewer #1, in addition to the new electrophysiology studies described above in revised Figure 4, as now shown in revised Figure 8A and B (see below), we performed additional experiments that characterized the abundance of surface AMPARs containing the GluA1 subunit along dendrites upon treatment with our KDAC inhibitors. Since expression of dendritic/synaptic Arc is well known to favor endocytosis of AMPARs (1-3), we reasoned that the increased abundance of Arc puncta caused by application of KDAC inhibitors, as shown in revised Figure 4C and D, should impact levels of surface AMPAR in a consistent manner. Excitingly, as predicted, treatment of primary cortical neurons with BDNF and with either one of the tested KDAC inhibitor caused a significant decrease in surface GluA1 as compared to control conditions (untreated and BDNF alone conditions). Importantly, these results are concordant with those collected in Figure 4C and D. Furthermore, to rule out that this change may be attributable to a reduction in *GluA1* mRNA levels, we performed a quantitative real-time PCR analysis and found no change in expression between key experimental conditions (Supplementary Figure 6).

We believe that these new results related to surface GluA1 expression and AMPAR-mediated synaptic transmission, now included in the revised manuscript, represents indirect *in vitro* evidence for an impact of enhanced Arc stability on synapse plasticity. We share the interest of the Reviewer to expand the scope of our work to understand the role of Arc acetylation in systems that are more complex and physiologically relevant. However, we believe that this question cannot be approach with just pharmacological strategies and that the development of new tools, like site-specific anti-acetyl-Arc antibodies, CRISPR/Cas genome editing constructs, and ultimately specific transgenic mouse models, will be required to satisfactorily pursue this area of research.

REVIEWER #3

Summary: In the article “Chemogenomic Analysis Reveals Key Role for Lysine Acetylation in Regulating Arc Stability”, Lalonde et. al. identify a set of small molecules that abrogate ubiquitin-mediated degradation of Arc. Further, they describe a mechanism where HDAC inhibition by the identified small molecules causes acetylation to accumulate on Arc lysines, which block the ability of these same lysines to be ubiquitinated. The authors rigorous experiments strongly support their conclusions. While there are certainly further questions to explore stemming from this work, sharing the present communication with the neurobiology field will stimulate the investigations of these new avenues. The manuscript is well written and I suggest only minor textual edits prior to publication in Nature Communications.

We humbly appreciate that Reviewer #3 recognizes the rigor of our experiments and suggests our manuscript worthy of publication in *Nature Communications* after only minor edits. Like the Reviewer, we acknowledge that many questions remain to be explored in relation with our findings and firmly believe that publication of our work would “stimulate the investigations of these new avenues”. In the revised version of our manuscript we have addressed the concerns raised by the Reviewer with revisions to the text that further clarify our results and methods.

1) In Figure 5D, I believe the MS/MS spectrum shows diglycine (G-G) as the lysine modification within the digested peptide sequence, not ubiquitin, as the G-G remnant remains after digestion of a ubiquitin-modified protein with trypsin (consistent with the workflow illustrated in Figure S7). The results perfectly support the claim that this is a ubiquitination site, but his subtle distinction should be explained in the figure legend and results section for clarity.

This was an oversight from our part and we thank the Reviewer for bringing this to our attention. Changes have been made to the Figure 5D, the text, and figure caption consistent with the Reviewer’s point.

2) The authors should clarify their comments of on Page 13, where they describe key proteomics results by stating that they detected peptides “... containing either an acetylation or ubiquitination mark at lysine residues 24, 33, and 55”. It is my interpretation that these three lysines were detected as sites of both acetylation and ubiquinylation, with each PTM detected in separate spectra. As it is currently stated, however, a reader could interpret it as some of these sites contained only one of the PTMs, but not both. This is another subtle distinction, but it is important for the mechanism suggested by the authors, where the sites are susceptible to both PTMs and acetylation can block ubiquitination of the same residues.

We take note of the Reviewer’s concern and made edits to the text that clarify this point.

3) I suggest some discussion on speculated mechanisms for Arc acetylation and ubiquitination. Previous work has implicated Triad3A in the ubiquitination of Arc (Neuron. 2014 Jun 18;82(6):1299-316.) It has also been previously shown that Arc interacts with the E3 ligase E6AP/UBE3A, but whether it is ubiquitinated by this enzyme has been questioned (Proc Natl

Acad Sci U S A. 2013 May 28;110(22):8888-93.). How does the current study fit into this body of literature? Similarly, could Arc's known interaction with the acetyltransferase TIP60 (eNeuro. 2014 Nov 12;1(1)) help explain the mechanism of its acetylation? Even if E6AP/UBE3A and TIP60 do not catalyze these PTMs on Arc, could their interactions suggest that Arc may have other interactors from these same classes of PTM-mediating enzymes? A bit of discussion here would help place the current findings into this previous body of literature.

We thank the Reviewer insightful comments and suggestions. We have discussed the interplay between different types of PTM above as well as cited very recent reports, which were not included in the first version of our manuscript, about Arc phosphorylation (5, 6) in our Discussion. We hope this Reviewer finds these sufficient and satisfactory at this point.

REVIEWER #4

Lalonde et al manuscript describing the role of lysine acetylation in regulating stability of Arc protein in neurons is beautifully written, all the experiments are done well for the most part, used solid statistical analyses to interpret data. There are no major technical concerns except for the overemphasis on the possibility of reversing Arc abnormalities to improve neurological impairments in different disease contexts. The significance of Arc as a therapeutic target needs to be substantiated with experimental evidence.

We thank the Reviewer for feedback regarding the overall quality of our manuscript and the studies described therein.

In terms of the perceived “...overemphasis on the possibility of reversing Arc abnormalities to improve neurological impairments in different disease contexts”, we sought in our writing and citation to be extremely careful to position our work strictly within the context of published evidence for dysregulated Arc expression in neurodevelopmental disorders (e.g. Angelman syndrome [7, 8]; Gordon Holmes syndrome [9]; fragile X syndrome [10, 11]; Alzheimer’s disease [12], Pitt-Hopkins syndrome, [13]; and schizophrenia [14-16]). We fully agree, however, that the significance of Arc modulation as a therapeutic target remains to be substantiated with experimental evidence that its dysregulation in these human disease contexts can have a disease modifying effect.

What we have tried to emphasize in our study is both: 1) a strategy (high-content imaging) to identify Arc modulators; and 2) a new facet of Arc biology (reversible lysine acetylation) that we show could be potentially targeted through pharmacological means to modulate Arc-dependent cellular mechanisms. In the revised version of the manuscript we have extended to correlations with effect on synaptic plasticity at the level of AMPAR density and electrophysiology. While we attempted to be careful so as not to over-interpret the therapeutic relevance of this findings, in light of the rapidly accumulating literature (for example see references 8 and 9) suggesting dysregulation of Arc as a component to the pathogenesis and pathophysiology of various brain disorders, we believe our work takes the important step forward of providing new pharmacological tools and a defined molecular process to ultimately test this hypothesis in relevant disease models.

The current manuscript indeed provides new insight into the biology of Arc protein in neurons. Title is very much appropriate for describing these findings. However the abstract, introduction and discussion sections of the manuscript emphasize on Arc as a therapeutic target. Along these lines, data is not convincing that Arc is the direct target of the lead compounds discovered by the chemogenomic screen. While this is a problem in general with phenotypic screens, additional experiments are required to demonstrate some sort of target engagement for the lead compounds.

Our chemogenomic study was designed to elucidate novel pathways involved in the intracellular regulation of Arc expression at the mRNA, protein, and subcellular trafficking in order to define specific molecular mechanisms controlling Arc function. In

this regard, our reference to Arc as “therapeutic target” is in the phenotypic sense and we agree that, while there can be significant advantages, a limitation initially in performing phenotypic screens is the need to elucidate the underlying molecular target(s) of a compound as compared to a direct biochemical screen assessing binding or target engagement as was performed in the elegant work of Worley and colleagues that led to the identification of the first reported Arc ligands (17).

Thus, we are not suggesting in the interpretation of our results that the three compounds (AK-7, oxamflatin, and CI-994 referred to as our lead compounds by the Reviewer) directly engage Arc protein. Rather, the direct biochemical targets of AK-7, oxamflatin, and CI-994 that we surmise are relevant to the changes in Arc stability as members of the lysine deacetylase (KDAC) family based upon the correlation of changes in Arc acetylation and histone acetylation. Based upon the known *in vitro* selectivity of the principal KDAC inhibitors we use, namely AK-7, an inhibitor of NAD⁺-dependent Class III lysine deacetylases in the sirtuin family, as well as CI-994 and oxamflatin, which share in common the inhibition of the zinc-dependent, Class I HDACs (HDAC1/2/3).

What are the phenotypes of Arc knockout and Arc overexpression in cultures? Could these compounds rescue the phenotype?

Rather than studying cellular phenotypes due to the overexpression or knocking out of Arc, which may be confounded by compensatory changes due to the relatively long term nature of such genetic experiments (relative to the time frame of a few hours in our study) and disruption of the physiologically relevant stoichiometry of Arc complexes, we designed our chemogenomic screen to harness the ability to quantitatively measure endogenous levels of Arc at the mRNA and protein level (i.e. not just mRNA level as more commonly done with reporters).

As such, our experimental system does not allow us to determine whether particular compounds tested in our chemogenomic screen could reverse phenotypes associated with Arc knockout since nuclear Arc protein abundance was used as the readout in our screen. In fact, by designing our screen this way we can only expect to detect pharmacological agents that act on Arc itself or the signaling supporting its nuclear and synaptic accumulation.

As for overexpression of Arc, several studies have already been described in the literature where, for instance, a reduction on GluA1 surface expression has been reported (e.g. see references 1 and 2). The fact that we found KDAC inhibitors can promote accumulation of Arc on one hand and reduce GluA1 surface expression on the other is consistent with these published observations.

Why was Neuro2A cells used for the experiment because the team is able to manipulate expression of Arc in neurons?

While we initially contemplated performing the mass spectrometry secondary assays using cultured primary cortical neurons, the ultimate success of our identification of novel

posttranslational modifications of Arc by mass spectrometry greatly depended on the amount of protein submitted for analysis because of the fact that the number of tryptic peptides with a specific modification over those that are unmodified can be greatly skewed in favor of the later in a sample. Hence, our strategy to maximize our chances to detect Arc tryptic peptides with a lysine acetylation modification evolved to be to overexpress tagged-Arc in Neuro2A cells in order to efficiently purify sufficient amount of Arc protein. Note that we were careful to demonstrate beforehand that co-application of AK-7 and oxamflatin to Neuro2A cell cultures resulted in greater abundance of total (both overexpressed and endogenous) and acetylated Arc protein (Figure 6B). This clear result suggested to us that the molecular machinery responsible for Arc acetylation was then conserved between mouse primary cortical neurons and Neuro2A cells.

Does the AK7 compound demonstrate any memory enhancing properties in their behavioral experiments?

As we mention in the final paragraph of our Discussion, at this point AK-7 has been found to limit neurodegeneration in mouse models of Huntington's, Parkinson's, and Alzheimer's diseases. As far as we know, the effect of AK-7 on learning and memory has not been directly investigated to date but we agree with the Reviewer that these would be exciting studies to perform in the future.

REFERENCES

1. S. Chowdhury, J. D. Shepherd, H. Okuno, G. Lyford, R. S. Petralia, N. Plath, D. Kuhl, R. L. Huganir, P. F. Worley, Arc/Arg3.1 interacts with the endocytic machinery to regulate AMPA receptor trafficking. *Neuron* **52**:445-459 (2006).
2. E. M. Rial Verde, J. Lee-Osbourne, P. F. Worley, R. Malinow, H. T. Cline, Increased expression of the immediate-early gene arc/arg3.1 reduces AMPA receptor-mediated synaptic transmission. *Neuron* **52**:461-474 (2006).
3. L. L. DaSilva, M. J. Wall, L. P de Almeida, S. C. Wauters, Y. C. Januário, J. Müller, S. A. Corrêa, Activity-regulated cytoskeleton-associated protein controls AMPAR endocytosis through a direct interaction with clathrin-adaptor protein 2. *eNeuro* pii: ENEURO.0144-15.2016 (2016).
4. C. L. Wee, S. Teo, N. E. Oey, G. D. Wright, H. M. VanDongen, A. M. VanDongen, Nuclear Arc interacts with the histone acetyltransferase Tip60 to modify H4K12 acetylation. *eNeuro* 1:pii: ENEURO.0019-14.2014. doi: 10.1523/ENEURO.0019-14.2014 (2014).
5. A. Gozdz, O. Nikolaienko, M. Urbanska, I. A. Cymerman, E. Sitkiewicz, M. Blazejczyk, M. Dadlez, C. R. Bramham, J. Jaworski, GSK3 α and GSK3 β phosphorylates Arc and regulate its degradation. *Front. Mol. Neurosci.* **10**:192. doi: 10.3389/fnmol.2017.00192 (2017).
6. O. Nikolaienko, M. S. Eriksen, S. Patil, H. Bito, C. R. Bramham, Stimulus evoked ERK-dependent phosphorylation of activity-regulated cytoskeleton-associated protein (Arc) regulates its neuronal subcellular localization. *Neuroscience* doi: <http://dx.doi.org/10.1016/j.neuroscience.2017.07.026> (2017).
7. P. L. Greer, R. Hanayama, B. L. Bloodgood, A. R. Mardinly, D. M. Lipton, S. W. Flavell, T. K. Kim, E. C. Griffith, Z. Waldon, R. Maehr, H. L. Ploegh, S. Chowdhury, P. F. Worley, J. Steen, M. E. Greenberg, The Angelman syndrome protein Ube3A regulates synapse development by ubiquitinating Arc. *Cell* **140**:704-716 (2010).
8. E. D. Pastuzyn, J. D. Shepherd, Activity-dependent Arc expression and homeostatic synaptic plasticity are altered in neurons from a mouse model of Angelman syndrome. *Front. Mol. Neurosci.* **10**:234. doi: 10.3389/fnmol.2017.00234 (2017).
9. N. Husain, Q. Yuan, Y. C. Yen, O. Pletnikova, D. Q. Sally, P. Worley, Z. Bichler, H. S. Je, TRIAD3A/RNF216 mutations associated with Gordon Holmes syndrome lead to synaptic and cognitive impairments via Arc misregulation. *Aging Cell* **16**:281-292 (2017).
10. S. Park, J. M. Park, S. Kim, J. A. Kim, J. D. Shepherd, C. L. Smith-Hicks, S. Chowdhury, W. Kaufmann, D. Kuhl, A. G. Ryazanov, R. L. Huganir, D. J. Linden, P. F. Worley, Elongation factor 2 and fragile X mental retardation protein control the dynamic translation of Arc/Arg3.1 essential for mGluR-LTD. *Neuron* **59**:70-83 (2008).

11. F. Niere, J. R. Wilkerson, K. M. Huber, Evidence for a fragile X mental retardation protein-mediated translational switch in metabotropic glutamate receptor-triggered Arc translation and long-term depression. *J. Neurosci.* **32**:5924-5936 (2012).
12. R. Koldamova, J. Schug, M. Lefterova, A. A. Cronican, N. F. Fitz, F. A. Davenport, A. Carter, E. L. Castranio, I. Lefterov, Genome-wide approaches reveal EGR1-controlled regulatory networks associated with neurodegeneration. *Neurobiol. Dis.* **63**:107-114 (2014).
13. A. J. Kennedy, E. J. Rahn, B. S. Paulukaitis, K. E. Savell, H. B. Kordasiewicz, J. Wang, J. W. Lewis, J. Posey, S. K. Strange, M. C. Guzman-Karlsson, S. E. Phillips, K. Decker, S. T. Motley, E. E. Swayze, D. J. Ecker, T. P. Michael, J. J. Day, J. D. Sweatt, Tcf4 regulates synaptic plasticity, DNA methylation, and memory function. *Cell Rep.* **16**:2666-2685 (2016).
14. M. Fromer, A. J. Pocklington, D. H. Kavanagh, H. J. Williams, S. Dwyer, P. Gormley, L. Georgieva, E. Rees, P. Palta, D. M. Ruderfer, N. Carrera, I. Humphreys, J. S. Johnson, P. Roussos, D. D. Barker, E. Banks, V. Milanova, S. G. Grant, E. Hannon, S. A. Rose, K. Chambert, M. Mahajan, E. M. Scolnick, J. L. Moran, G. Kirov, A. Palotie, S. A. McCarroll, P. Holmans, P. Sklar, M. J. Owen, S. M. Purcell, M. C. O'Donovan, De novo mutations in schizophrenia implicate synaptic networks. *Nature* **506**:179-184 (2014).
15. S. M. Purcell, J. L. Moran, M. Fromer, D. Ruderfer, N. Solovieff, P. Roussos, C. O'Dushlaine, K. Chambert, S. E. Bergen, A. Kähler, L. Duncan, E. Stahl, G. Genovese, E. Fernández, M. O. Collins, N. H. Komiyama, J. S. Choudhary, P. K. Magnusson, E. Banks, K. Shakir, K. Garimella, T. Fennell, M. DePristo, S. G. Grant, S. J. Haggarty, S. Gabriel, E. M. Scolnick, E. S. Lander, C. M. Hultman, P. F. Sullivan, S. A. McCarroll, P. Sklar, A polygenic burden of rare disruptive mutations in schizophrenia. *Nature* **506**:185-190 (2014).
16. F. Managò, M. Mereu, S. Mastwal, R. Mastrogiacomo, D. Scheggia, M. Emanuele, M. A. De Luca, D. R. Weinberger, K. H. Wang, F. Papaleo, Genetic disruption of Arc/Arg3.1 in mice causes alterations in dopamine and neurobehavioral phenotypes related to schizophrenia. *Cell Rep.* **16**:2116-2128 (2016).
17. W. Zhang, J. Wu, M. D. Ward, S. Yang, Y. A. Chuang, M. Xiao, R. Li, D. J. Leahy, P. F. Worley, Structural basis of Arc binding to synaptic proteins: Implications for cognitive disease. *Neuron* **86**:490-500 (2015).

Reviewers' Comments:

Reviewer #1 (Remarks to the Author):

The authors have addressed my concerns.

Reviewer #2 (Remarks to the Author):

The authors have addressed all the important points raised. The authors have added new experiments to address the importance of Arc acetylation in primary neurons by performing electrophysiology and immunocytochemistry.

I have no further reservations in the publication of this manuscript.

Reviewer #3 (Remarks to the Author):

Lalonde and colleagues have made significant improvements in the content and clarity of their manuscript describing the regulation of Arc protein stability by acetylation. I believe that the neurobiology field will find this an impactful contribution to the literature--I believe that the data warrants publication in Nature Communications, with only any remaining minor textual revisions needed beforehand. On this note, this reviewer suggests subtle changes to Figure 6C: The header "Number of Times Identified" should be changed to "Peptide Spectral Counts", as PSMs are a standard unit of measure in the proteomics field that applies here. Also, I question whether the "Site Number" column on the left is needed, as this is just a count of the number of rows in the table (ie., counting the sites 1 to 6 from N to C term seems a bit arbitrary)--and a reader could incorrectly interpret this as a residue position on a peptide or protein (which is indicated in the next column over).